# Ice Cream Doesn't Cause Drowning: Benchmarking LLMs Against Statistical Pitfalls in Causal Inference

**Jin Du**[1]**, Li Chen**[1]**, Xun Xian**[1]**, An Luo**[1]**, Fangqiao Tian**[1]**, Ganghua Wang**[2]**,**
**Charles Doss**[1]**, Xiaotong Shen**[1]**, Jie Ding**[1]

[1]School of Statistics, University of Minnesota, Minneapolis, MN 55455
[2]Data Science Institute, University of Chicago, Chicago, IL 60637
`{du000142,chen7019,xian0044,luo00318,tian0257}@umn.edu`
`{cdoss,xshen,dingj}@umn.edu, ganghua@uchicago.edu`

## Abstract

Reliable causal inference is essential for making decisions in high-stakes areas like medicine, economics, and public policy. However, it remains unclear whether large language models (LLMs) can handle rigorous and trustworthy *statistical causal inference*. Current benchmarks usually involve simplified tasks. For example, these tasks might only ask LLMs to identify semantic causal relationships or draw conclusions directly from raw data. As a result, models may overlook important statistical pitfalls, such as Simpson's paradox or selection bias. This oversight limits the applicability of LLMs in the real world. To address these limitations, we propose **CausalPitfalls**, a comprehensive benchmark designed to rigorously evaluate the capability of LLMs in overcoming common causal inference pitfalls. Our benchmark features structured challenges across multiple difficulty levels, each paired with grading rubrics. This approach allows us to quantitatively measure both causal reasoning capabilities and the reliability of LLMs' responses. We evaluate models using two protocols: (1) direct prompting, which assesses intrinsic causal reasoning, and (2) code-assisted prompting, where models generate executable code for statistical analysis. Additionally, we validate the effectiveness of this judge by comparing its scoring with assessments from human experts. Our results reveal significant limitations in current LLMs when performing statistical causal inference. The CausalPitfalls benchmark provides essential guidance and quantitative metrics to advance the development of trustworthy causal reasoning systems. Our code is publicly available at [CausalPitfalls](#).

## 1 Introduction

Causal inference (Pearl, 2009; Imbens & Rubin, 2015) is fundamental to decision-making across diverse fields. For instance, accurately determining the effectiveness and safety of a vaccine is pivotal in public health decisions (Voysey et al., 2021). However, identifying causal relationships with both reliability and interpretability remains challenging. In practice, individuals without formal statistical training frequently fall into subtle pitfalls, leading to plausible yet incorrect conclusions. A classic illustration is the erroneous conclusion that ice cream sales cause drowning incidents — overlooking the hidden confounder of hot weather causing both events (Pearl, 2009; Greenland & Robins, 1986; Rosenbaum, 1987).

Given these complexities, automated tools like large language models (LLMs) present promising avenues, demonstrated by their effectiveness in scientific problem-solving (Lewkowycz et al., 2022; Achiam et al., 2023; Lin et al., 2025a;b) and clinical reasoning (Singhal et al., 2023; Yu et al., 2025). Recent studies (Wang, 2024; Dhawan et al., 2024; Liu et al., 2024) have evaluated LLMs' abilities to evaluate accuracy in causal-effect estimation, but these benchmarks often neglect crucial aspects like robustness, interpretability, and susceptibility to common causal pitfalls. As a result, LLMs can produce seemingly convincing yet misleading causal conclusions.

To illustrate why reliability assessment is crucial, we highlight two representative failure modes (detailed in Section 3). First, LLMs can ignore strong data evidence, but in favor of superficial semantic cues: in a synthetic health scenario with identical datasets, LLMs concluded the drink was beneficial when labeled "HealthPlus" and harmful when labeled "UltraSugar," even when the data indicated the **opposite**. Second, LLMs can mistake random variation for genuine causal structure: when tested on research funding data from the Netherlands, LLMs attributed differences in success rates to gender bias or Simpson's paradox, despite statistical analyses showing that neither claim is supported. These cases demonstrate that LLMs may produce confident causal claims directly contradicted by the data, showing the need for benchmarks that assess causal inference *reliability*.

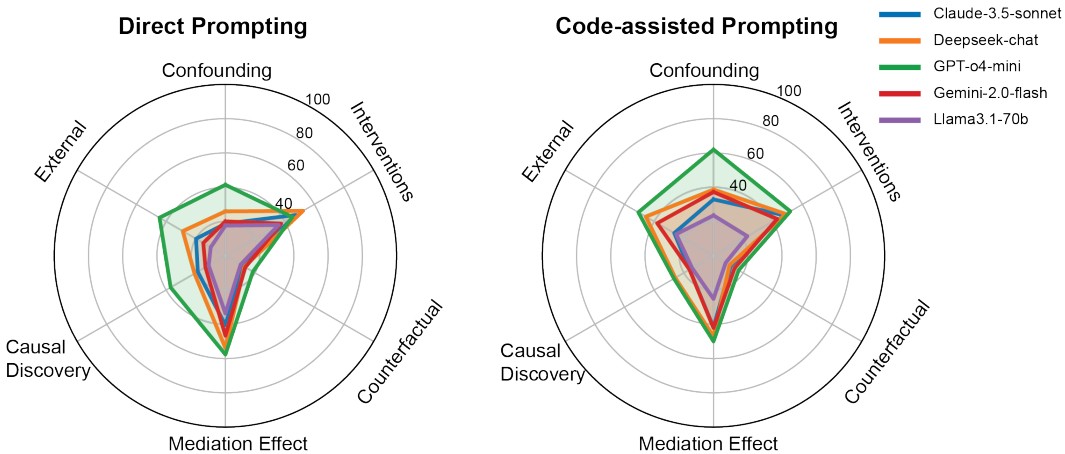

Figure 1: **Overall Message:** Our results reveal a clear **reliability gap** in causal inference when LLMs rely only on direct prompting, with all models struggling most on mediation and external validity questions. Introducing code-assisted prompting leads to substantial gains across every task and brings all models closer together in performance. This shows that executable analysis is essential for large language models to handle complex statistical challenges and deliver trustworthy causal conclusions. Full results for all evaluated LLMs are provided in Table 4.

## 1.1 MAIN CONTRIBUTIONS

First, we introduce **CausalPitfalls**, a novel comprehensive benchmark specifically designed to evaluate the reliability of large language models (LLMs) in statistical causal inference. Unlike existing benchmarks primarily focused on accuracy, our benchmark targets model susceptibility to common causal pitfalls as shown in Figure 1, including (1) Confounding Biases and Spurious Associations, (2) Interventions and Experimental Reasoning, (3) Counterfactual Reasoning and Hypotheticals, (4) Mediation and Indirect Causal Effects, (5) Causal Discovery and Structure Learning, and (6) Causal Generalization and External Validity (Pearl et al., 2003; Peters et al., 2017). These categories are structured into 15 distinct challenges, encompassing a total of 75 evaluation questions and 75 carefully constructed datasets that systematically test the robustness of LLM causal reasoning capabilities.

Second, we comprehensively evaluate the reliability of ten LLMs under two distinct evaluation protocols: (1) *direct prompting*, assessing intrinsic causal reasoning from raw data, and (2) *code-assisted prompting*, where models generate executable code to perform statistical analyses before responding. This dual-protocol approach provides a detailed quantitative assessment, highlighting areas where computational assistance significantly improves causal reasoning and where intuitive reasoning could suffice.

Third, we introduce a quantitative metric termed *causal reliability*, calculated as the average normalized score across all benchmark challenges, enabling standardized comparisons of LLM reliability in causal reasoning tasks. By systematically quantifying reliability, this metric provides a crucial framework for future research aimed at developing more robust and trustworthy causal inference capabilities in AI systems.

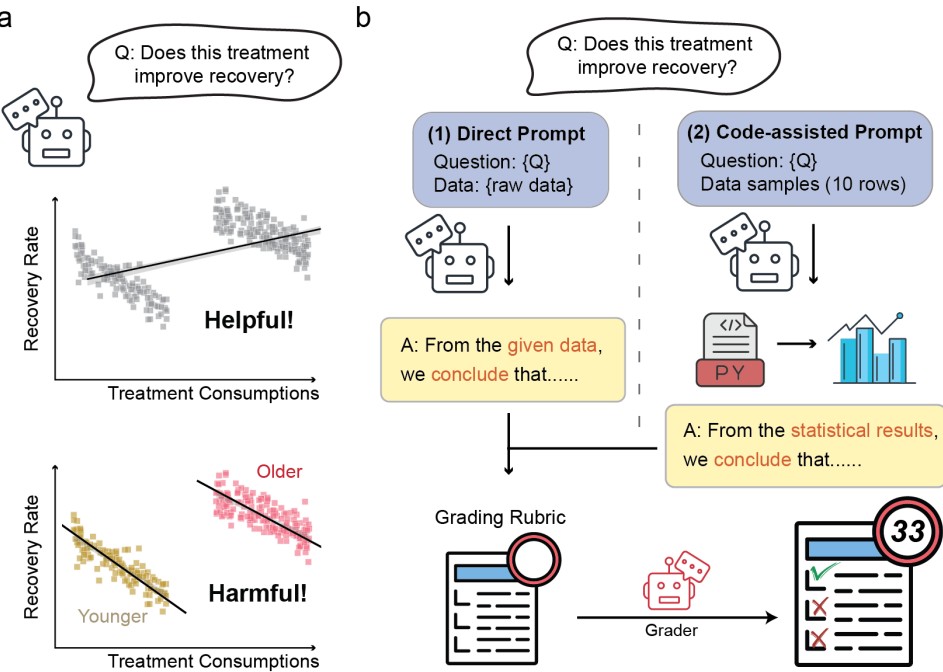

Figure 2: High-level overview of the CausalPitfalls benchmark. (a) An illustrative real-world pitfall (Simpson's paradox): when data on treatment consumption and recovery are pooled (top), a naïve analysis finds a positive effect ("Helpful!"), but stratifying by age reveals a negative effect within both younger and older subgroups ("Harmful!"). (b) Benchmark workflow: LLMs are evaluated under two protocols: (1) Direct Prompting on raw data, assessing intrinsic causal reasoning, and (2) Code-Assisted Prompting on sampled data, assessing computationally grounded inference. In both cases, model answers are automatically scored against a hidden grading rubric by an independent grader to quantify each model's causal reliability.

## 1.2 RELATED WORK

**1. Causal Inference and Statistical Pitfalls.** Causal inference from observational data is inherently challenging because counterfactuals are unobservable and confounding is ubiquitous (Pearl, 2009; Imbens & Rubin, 2015). Causal inference methods for addressing confounders, whether the confounders are measured (Chan et al., 2016; Lin et al., 2023; Doss et al., 2024) or latent (Kang et al., 2016; Guo et al., 2022), depend on restrictive model assumptions that are often difficult to verify empirically. Inferring causal direction similarly hinges on stringent structural equation assumptions (Peters & Bühlmann, 2014; Li et al., 2024) or auxiliary information such as valid instruments (Chen et al., 2024). Mediation analysis (MacKinnon, 2012; Yang et al., 2021), which targets specific causal pathways, demands careful adjustment for intermediate variables to avoid post-treatment bias. Finally, transporting causal conclusions across different domains requires justification of source-target invariances and methodologies for causal knowledge transfer (Wei et al., 2023; Chen et al., 2025). Rigorously confronting each of these challenges is essential to conduct a reliable causal analysis.

**2. LLMs for Causal Reasoning.** Recent studies have extensively investigated the causal reasoning capabilities of LLMs (Willig et al., 2022; Zečević et al., 2023; Qin et al., 2019; Luo et al., 2025; Du et al., 2024). For example, Kiciman et al. (2023) demonstrated that LLMs can infer causal relationships only from variable names, outperforming traditional statistical approaches (Peters et al., 2017). However, these evaluations focus on scenarios involving commonsense causality. To bridge this gap, Jin et al. (2023) introduced synthetic datasets generated from causal graphs, thereby enabling the assessment of LLMs' causal reasoning performance in contexts extending beyond commonsense knowledge. Additionally, recent works (Long et al., 2023; Zhou et al., 2024) have assessed LLMs' capabilities in data-driven causal inference tasks, focusing on accuracy in estimating causal effects and recovering DAG structures from observational data.

## 2 BENCHMARK CURATION

### 2.1 PITFALL CATEGORIES AND CHALLENGES

To evaluate the reliability of causal inference performed by LLMs, we introduce the benchmark **CausalPitfalls** to assess model performance across common statistical pitfalls. Specifically, our benchmark addresses six major categories of causal inference pitfalls, consisting of 15 distinct challenges. Each challenge includes five questions across difficulty levels ranging from "very easy" to "very hard." Table 1 summarizes the categories and their respective challenges:

Table 1: CausalPitfalls benchmark categories and challenges

| Confounding biases and spurious associations | Interventions and experimental reasoning |
|---|---|
| Simpson's paradox | Observational vs experimental reasoning |
| Selection bias (Berkson's paradox) | Causal effect estimation |
| **Counterfactual reasoning and hypotheticals** | **Mediation and indirect causal effects** |
| Counterfactual outcome prediction | Mediator-outcome confounding |
| Causal necessity and sufficiency | Sequential mediators |
| | Treatment-mediator interaction effects |
| **Causal discovery and structure learning** | **Causal generalization and external validity** |
| Cause-effect direction inference | Population shift and transferability |
| Handling uncertainty in causal structures | Temporal stability of causal effects |
| | Contextual interaction and moderation effects |
| | Domain shift and transportability of causal knowledge |

Below is a brief overview of the six major categories in the **CausalPitfalls** benchmark:

- **Confounding biases and spurious associations:** Covers scenarios where misleading correlations arise due to hidden variables or biased conditioning.
- **Interventions and experimental reasoning:** Focuses on distinguishing correlation from causation through randomized experiments or interventional data, and quantifying treatment effects.
- **Counterfactual reasoning and hypotheticals:** Evaluates LLMs' ability to answer "what if" questions by reasoning about alternate outcomes under hypothetical changes.
- **Mediation and indirect causal effects:** Assesses whether models can identify and reason through intermediate causal pathways, including complex mediation structures.
- **Causal discovery and structure learning:** Tests the ability to infer causal directionality or relationships from data without pre-defined causal graphs.
- **Causal generalization and external validity:** Examines whether causal conclusions generalize across different contexts, populations, and environments.

Many of these scenarios can also be framed as purely statistical association problems. However, throughout this benchmark, we adopt a causal interpretation grounded in either the Neyman–Rubin potential outcomes framework (Splawa-Neyman et al., 1990; Rubin, 1974) or Pearl's do-calculus (Pearl, 2009). This distinction ensures that our tasks target causal reasoning, rather than surface-level correlations.

One illustrative example of these pitfalls is Simpson's paradox (Simpson, 1951) (Fig. 2a), a commonly misunderstood statistical phenomenon. Simpson's paradox occurs when overall data seem to suggest one conclusion: for example, patients recover better with a particular medical treatment. At first glance, the treatment might appear beneficial. However, once the patients are divided into subgroups based on age, the same data show that the treatment is harmful within each age group. An incorrect analysis would overlook the importance of these subgroups, mistakenly suggesting the treatment is effective. Importantly, the stratification variable in Simpson's paradox is assumed to be a confounder, not a collider.

### 2.2 EVALUATION PROTOCOLS AND DATA

**LLM-based Causal Inference Protocols.** In this paper, we consider two unified protocols to evaluate LLM reliability (Fig. 2b): **(1) Direct Prompting**: LLMs directly answer causal inference

Table 2: Questions by Difficulty Level for "Simpson's Paradox"

| Difficulty | Question |
|---|---|
| Very Easy | Evaluate whether {TREATMENT} causally affects {OUTCOME}. Adjust for the known confounder ({CONFOUNDER}) using stratification or regression. State if Simpson's paradox is present, and provide adjusted rates with 95% confidence intervals along with a recommendation. |
| Easy | Evaluate whether {TREATMENT} causally affects {OUTCOME}. Consider the potential influence of the confounder ({CONFOUNDER}), adjust accordingly, and assess Simpson's paradox. Provide adjusted effect sizes with uncertainty estimates and a recommendation. |
| Medium | Evaluate the causal impact of {TREATMENT} on {OUTCOME}. Account for relevant confounders. Provide adjusted effects with uncertainty measures and justify your recommendation. |
| Hard | Assess the causal relationship between {TREATMENT} and {OUTCOME}, considering potential confounders. |
| Very Hard | Evaluate whether {TREATMENT} causally affects {OUTCOME} without additional hints. |

questions based on the provided raw data. This approach tests the models' intrinsic capability to perform causal inference without additional computational tools or external support. **(2) Code-Assisted Prompting**: LLMs generate executable code to perform statistical analysis relevant to the questions, then interpret the results to answer the questions. This method assesses the LLMs' ability to translate causal reasoning tasks into accurate computational procedures and use analytical results to avoid common pitfalls.

**Questions.** Each challenge includes five versions of the core question, ranging from very easy to very hard (Table 2). The easier versions give the model more guidance. For example, pointing out the confounder to adjust for or asking directly about Simpson's paradox. As the difficulty increases, these hints are gradually removed. This setup lets us test whether models can still recognize and handle causal pitfalls when less direction is given (Fig. 2b).

**Datasets.** To construct datasets tailored to each challenge, we utilize causal graphs following Pearl et al. (2003) and Peters et al. (2017). For every statistical pitfall, we select causal graphs that capture its unique complexities and characteristics. Each challenge is accompanied by five distinct datasets, each containing over 500 samples for comprehensive evaluation. Our simulation approach uses structural causal models based on directed acyclic graphs (DAGs), where each structural equation represents a causal mechanism rather than merely a statistical association. The coefficients in these equations directly encode the causal effects, allowing us to define the ground truth against which inference methods can be evaluated. This approach is mathematically equivalent to simulating potential outcomes under the specified causal structure. The structural equations include both linear and non-linear forms (e.g., non-linear link functions and interaction terms), so the evaluation is not limited to purely linear relationships.

## 2.3 EVALUATION METRICS

To evaluate the reliability of LLMs for causal inference, we developed detailed grading rubrics for each causal pitfall, informed by guidelines from Sterne et al. (2016); Vandenbrouckel et al. (2007). Each benchmark challenge includes multiple questions, each assigned points based on how effectively the model addresses the specific pitfall (see Appendix for detailed rubric). The total *score* for a challenge is the sum of points obtained across these questions, and *max_score* is the maximum achievable score. To enable fair comparisons across challenges, we compute a normalized score:

$$\text{Normalized Score (\%)} = \frac{\text{score}}{\text{max\_score}} \times 100\%. \tag{1}$$

We evaluate LLM responses automatically using an independent GPT-4o model (Achiam et al., 2023) to minimize potential biases. To validate the accuracy of this automated evaluation, we additionally engaged three statisticians to manually grade 150 randomly selected responses. We

measure consistency between automated and human scores using the *gap* metric:

$$\text{Gap} = \frac{1}{150} \sum_{i=1}^{150} \frac{|\text{score}_{\text{LLM}}^{(i)} - \text{score}_{\text{human}}^{(i)}|}{s_{\text{max},i}} \in [0, 1],$$

where $s_{\text{max},i}$ is the maximum score of corresponding challenge, and $\text{score}_{\text{LLM}}^{(i)}$, $\text{score}_{\text{human}}^{(i)} \in \mathbb{N}^+$ are scores from automated and human evaluations, respectively. Here, the gap metric ranges from 0 to 1, where 0 indicates perfect agreement.

Finally, to provide a summary metric, we define *causal reliability* as the average normalized score across all benchmark challenges. This measure captures the overall trustworthiness and reliability of LLMs in statistical causal inference tasks.

## 3 ILLUSTRATIVE PITFALLS IN CAUSAL REASONING

When evaluating causal inference with LLMs, surface-level answers can create an illusion of competence. Statistical causal inference requires grounding conclusions in evidence, checking assumptions, and ruling out alternatives. LLMs, however, may produce confident but flawed outputs that rely on irrelevant cues or statistical artifacts, giving a false sense of robustness.

We illustrate this problem with two failure cases. The first shows how models can base conclusions on superficial semantic cues instead of data, while the second shows how they may misinterpret random variation as causal structure.

**Branding Bias: Adversarial Sensitivity to Branding and Semantic Manipulation.** As an illustrative example, we examined whether LLMs can be misled by superficial cues when drawing causal conclusions. We constructed a synthetic scenario in which beverage consumption affected health outcomes. In the setting, other factors, lifestyle and health awareness, acted as confounders, but the brand label itself had no causal role (Fig. 3).

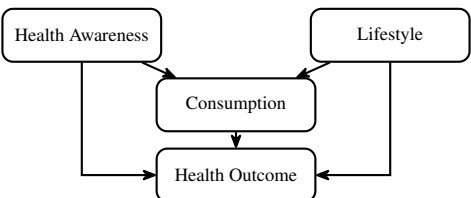

Figure 3: Causal DAG illustrating how beverage consumption, health awareness, and lifestyle affect health outcomes. The beverage's brand name ("HealthPlus" or "UltraSugar") does not causally influence outcomes.

Despite identical data, simply changing the brand name from a healthy-sounding label (*HealthPlus*) to a harmful-sounding one (*UltraSugar*) induced changes in LLMs' conclusions (Table 3). GPT-4o and Gemini-2.0-flash, for example, made a conclusion aligning purely with the brand semantics, rather than the provided data.

The branding bias example shows that LLMs may rely on superficial semantic cues, attributing causal effects to labels even when the underlying data provide evidence **to the contrary**.

Table 3: Branding bias evaluation. Each row represents a combination of the beverage's label ("HealthPlus" or "UltraSugar") and the true effect of the given data (beneficial or harmful). A checkmark (✓) indicates a beneficial conclusion and a cross (×) a harmful conclusion; correct inferences are those that match the true effect.

| Brand Label | True Effect (Data) | GPT-4o | Gemini-2.0-flash | Claude-3.5-sonnet |
|---|---|---|---|---|
| HealthPlus | ✓ | ✓ | ✓ | ✓ |
| UltraSugar | ✓ | × | × | ✓ |
| HealthPlus | × | ✓ | ✓ | ✓ |
| UltraSugar | × | × | × | ✓ |

**Spurious Causal Inference from Random Patterns.** As a second illustrative example, we show how LLMs can mistake random variation in real-world data for genuine causal structure. Specifically, we evaluated LLMs on real data from a PNAS study (Van der Lee & Ellemers, 2015) that analyzed funding success rates across academic disciplines in the Netherlands. When asked if the data reveal gender bias favoring male applicants, all of the tested LLMs drew incorrect conclusions, either attributing the differences in raw percentages directly to gender or considering Simpson's paradox.

However, a careful statistical analysis as illustrated in the Appendix following Irizarry (2019) shows that women are not disproportionately applying to more competitive disciplines, so Simpson's paradox does not apply. Furthermore, computing log-odds ratios divided by their standard errors across disciplines and adjusting for multiple comparisons reveals no statistical evidence of gender bias within any department. None of the tested LLMs recognized these key insights.

The spurious inference example shows that LLMs may mistake random variation in observational data for genuine causal structure, failing to apply the statistical checks needed to rule out noise.

## 4 RESULTS AND ANALYSIS

In this section, we present the key findings from our experiments, examining the causal inference capabilities of ten large language models. Our evaluation spans both closed source and open source LLMs, with the complete list of models provided in the Appendix. All models were evaluated using two protocols: direct prompting and code assisted prompting. The assessment covered six categories of causal inference pitfalls and fifteen challenges, each built from five datasets and five questions of varying difficulty.

Table 4: Causal reliability across causal pitfalls, comparing direct and code-assisted prompting. Values represent averages of normalized scores, defined in equation (1), across five questions per pitfall category; higher scores indicate better performance.

| LLM (Direct Prompting) | Conf | Interv | Counter | Med | Disc | Ext | Average |
|---|---|---|---|---|---|---|---|
| Gemma2-9b | 14.00 | 30.72 | 8.00 | 15.89 | 9.49 | 4.05 | 13.69 |
| Llama3.1-8b | 18.46 | 34.78 | 10.86 | 29.33 | 7.71 | 6.05 | 17.86 |
| Llama3.1-70b | 17.60 | 35.92 | 10.29 | 33.67 | 11.31 | 9.92 | 19.78 |
| Mistral-7b | 17.31 | 29.83 | 5.71 | 19.17 | 8.40 | 6.18 | 14.43 |
| Mixtral-8x22b | 16.40 | 32.08 | 8.00 | 27.50 | 9.20 | 8.75 | 16.99 |
| Claude-3.5-sonnet | 18.74 | 47.60 | 12.00 | 40.50 | 18.63 | 19.84 | 26.22 |
| Gemini-2.0-flash | 20.06 | 37.56 | 13.43 | 46.67 | 13.48 | 14.93 | 24.36 |
| Deepseek-chat | 25.89 | **52.42** | 12.86 | 53.83 | 20.83 | 28.74 | 32.43 |
| GPT-4.1 | 17.26 | 33.57 | 6.57 | 53.27 | 16.43 | 24.31 | 25.24 |
| GPT-o4-mini | **41.43** | 45.21 | **18.57** | 57.67 | **36.97** | **44.48** | **40.72** |

| LLM (Code-Assisted Prompting) | Conf | Interv | Counter | Med | Disc | Ext | Average |
|---|---|---|---|---|---|---|---|
| Gemma2-9b | 7.89 | 18.50 | 4.86 | 23.04 | 6.63 | 18.50 | 13.24 |
| Llama3.1-8b | 11.93 | 15.99 | 7.67 | 18.06 | 6.23 | 17.29 | 12.86 |
| Llama3.1-70b | 23.54 | 22.83 | 8.21 | 25.17 | 14.41 | 25.17 | 19.89 |
| Mistral-7b | 4.68 | 13.15 | 1.43 | 11.11 | 6.43 | 9.09 | 7.65 |
| Mixtral-8x22b | 22.50 | 30.53 | 5.44 | 26.49 | 14.29 | 24.31 | 20.59 |
| Claude-3.5-sonnet | 32.91 | 47.33 | 11.71 | 41.73 | 16.12 | 26.47 | 29.38 |
| Gemini-2.0-flash | 37.20 | 42.96 | 14.29 | 42.17 | 16.19 | 37.98 | 31.80 |
| Deepseek-chat | 38.63 | 48.70 | 10.86 | 47.13 | 25.79 | 45.63 | 36.12 |
| GPT-4.1 | 47.14 | 42.65 | 12.29 | 49.40 | 23.87 | 48.58 | 37.32 |
| GPT-o4-mini | **62.00** | **51.86** | **16.96** | 50.00 | 26.67 | 50.71 | **43.03** |

*Conf: Confounding biases and spurious associations; Interv: Interventions and experimental reasoning; Counter: Counterfactual reasoning and hypotheticals; Med: Mediation and indirect causal effects; Disc: Causal discovery and structure learning; Ext: Causal generalization and external validity.*

**Overall Performance.** Across models, **GPT-o4-mini** demonstrated the highest overall reliability, achieving an average causal reliability of 40.72% under direct prompting and 43.03% under code-

assisted prompting (Table 4). Deepseek-chat, although lower in aggregate (average 32.43% under direct prompting and 36.12% under code-assisted prompting), obtained the strongest performance in *interventions and experimental reasoning* (52.42% under direct prompting and 48.70% under code-assisted prompting). Taken together, these results indicate that optimized mid-scale models can, in certain contexts, outperform larger frontier systems on causal reasoning tasks.

**Benefits of Code-Assisted Prompting.** Although code-assisted prompting improved several strong models, its benefits were not universal. For example, GPT-o4-mini and Gemini-2.0-flash gained from code execution, increasing their averages from 40.72% to 43.03% and 24.36% to 31.80%, respectively. GPT-4.1 also improved significantly (25.24% $\rightarrow$ 37.32%). Under code-assisted prompting, models first convert raw data into summary statistics through generated code, then reason over those statistics. This separates low-level data parsing from causal reasoning, which partly explains why stronger models benefit more: they produce correct analysis code and reason on clean numerical outputs rather than raw tables. In contrast, some small open-source LLMs showed little benefit: Llama3.1-8B decreased from 17.86% to 12.86%, Gemma2-9B remained unchanged (13.69% to 13.24%), and Mistral-7B even dropped from 14.43% to 7.65%. These results indicate that computational results increase the strengths of already-capable LLMs, but does not provide a uniform advantage across the LLMs. Models with high code-error rates (e.g., Mistral-7B, Llama-8B) have worse scores under code-assisted prompting. Allowing one debugging attempt can bring their performance back to direct-prompting levels. We discuss this further in Appendix H.

**Impact of Difficulty Levels.** Causal reliability consistently decreases as the question becomes harder. As shown in Table 5, all models performed best on very easy and easy questions, with average scores declining as task difficulty increased, where the question contains fewer hints (see Table 2. For example, GPT-o4-mini achieved 60.72% under direct prompting and 56.73% under code-assisted prompting on very easy items, but dropped to 17.75% and 32.84%, respectively, on very hard questions. Deepseek-chat and Gemini-2.0-flash showed similar trends, maintaining moderate performance on medium and hard levels but falling quickly on the *very hard* questions. These results indicate that code assistance is particularly beneficial for challenging tasks, helping stronger models recover some performance at higher difficulty levels.

Table 5: Causal reliability by difficulty levels of questions, comparing direct and code-assisted prompting. Values represent averages of normalized score, defined in equation (1), across 15 challenges; higher scores indicate better performance.

| **LLM (Direct Prompting)** | Very Easy | Easy | Medium | Hard | Very Hard |
|---|---|---|---|---|---|
| Gemma2-9b | 20.50 | 15.64 | 14.22 | 6.72 | 5.46 |
| Llama3.1-8b | 27.99 | 20.85 | 19.90 | 10.71 | 5.82 |
| Llama3.1-70b | 31.04 | 27.38 | 22.04 | 11.29 | 5.23 |
| Mistral-7b | 20.04 | 18.88 | 18.02 | 7.49 | 3.80 |
| Mixtral-8x22b | 27.80 | 21.20 | 20.31 | 8.41 | 5.23 |
| Claude-3.5-sonnet | 40.27 | 36.63 | 29.69 | 13.42 | 8.30 |
| Gemini-2.0-flash | 37.83 | 33.59 | 27.59 | 15.72 | 8.22 |
| Deepseek-chat | 46.37 | 38.92 | 37.92 | 28.79 | 16.36 |
| GPT-4.1 | 38.01 | 33.94 | 33.59 | 17.31 | 7.67 |
| GPT-o4-mini | **60.72** | **54.05** | **48.13** | **30.72** | **17.75** |

| **LLM (Code-Assisted Prompting)** | Very Easy | Easy | Medium | Hard | Very Hard |
|---|---|---|---|---|---|
| Gemma2-9b | 19.09 | 18.35 | 16.43 | 10.56 | 7.94 |
| Llama3.1-8b | 26.04 | 14.41 | 12.77 | 8.84 | 7.90 |
| Llama3.1-70b | 32.21 | 27.88 | 24.77 | 14.08 | 7.58 |
| Mistral-7b | 9.44 | 12.03 | 9.59 | 6.63 | 2.73 |
| Mixtral-8x22b | 33.14 | 29.77 | 25.64 | 14.59 | 7.77 |
| Claude-3.5-sonnet | 39.70 | 35.06 | 33.55 | 20.92 | 18.34 |
| Gemini-2.0-flash | 46.01 | 39.83 | 32.17 | 28.00 | 21.67 |
| Deepseek-chat | 55.56 | 48.23 | 43.69 | 28.46 | 16.50 |
| GPT-4.1 | **57.80** | 46.49 | 46.48 | 24.33 | 19.65 |
| GPT-o4-mini | 56.73 | **49.66** | **49.37** | **34.31** | **32.84** |

**Persistent Reliability Gaps.**    Despite variation across models and prompting strategies, significant reliability gaps remain in all settings. Among the tested LLMs, GPT-o4-mini achieved the highest causal reliability score $43.03\%$ on average under code-assisted prompting, while most other models remained well below this level. Performance on difficult causal reasoning challenges was especially limited, with *very hard* questions rarely exceeding $30\%$ even for the best LLMs. Taken together, the results suggest that existing LLMs, when used without finetuning or specialized architectures, remain unreliable to apply in high-stakes causal inference.

**Human-LLM Grading Alignment.**    To validate the fidelity of our automated GPT-4o scoring against expert judgments, we conducted a human validation study on a stratified sample of 150 model responses, with equal representation across the six pitfall categories and five difficulty levels (see Appendix). Three PhD students in statistics independently graded each response using our detailed rubrics. We then compared the resulting human scores with those produced by GPT-4o via the gap metric, which yielded a mean value of 0.11. This close agreement confirms that our GPT-4o evaluator reliably mirrors expert assessments, justifying its use for large-scale, reproducible evaluation of LLM performance in causal inference without the need for extensive human oversight.

**Code-Assisted Execution Errors.**    As shown in Figure 4, code-execution failures peak in the "mediation effects" and "interventions and experimental reasoning" categories, where implementing correct stratification and transportability routines is most demanding. Interestingly, "very easy" questions produce the highest failure rates, whereas "very hard" questions yield lower rates.

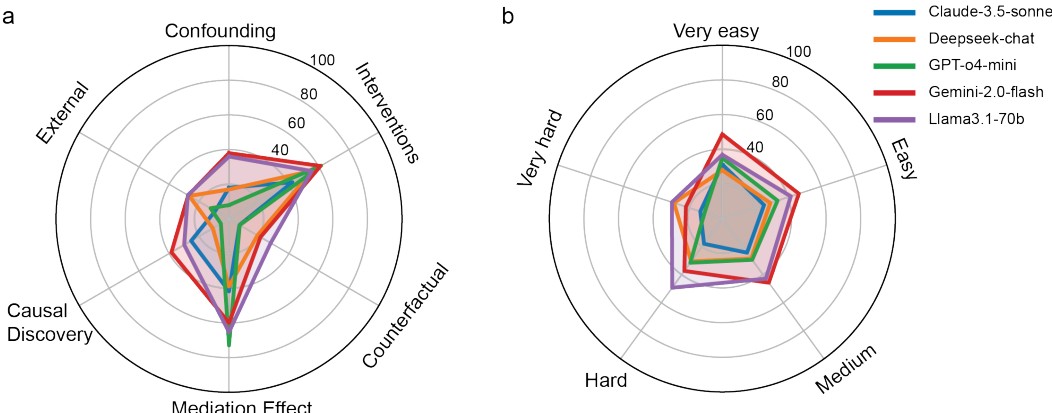

Figure 4: **Code execution failure rates (%) in code-assisted prompting protocol across causal inference challenges and question difficulty.** Failure rate is defined as the percentage of code-generation attempts that either raise execution errors or produce invalid analytical outputs, computed only for the code-assisted prompting protocol. (a) Average failure rate for each of the six causal-inference pitfall categories. (b) Average failure rate by question difficulty level, increasing from very easy through very hard tasks.

## 5    CONCLUSION

We introduced **CausalPitfalls**, a benchmark designed to rigorously evaluate the reliability of LLMs in performing statistical causal inference. Unlike existing benchmarks that focus primarily on accuracy, our benchmark reveals how LLMs can produce confident yet flawed conclusions by falling into classical statistical pitfalls. Our results indicate substantial gaps in reliability across all models and settings. Even state-of-the-art models exhibit systematic vulnerabilities to confounding, semantic bias, and difficulties in generalizing causal knowledge across contexts. These findings highlight an urgent need for targeted interventions to improve LLMs' trustworthiness in scientific and policy domains. Future directions include expanding the benchmark to cover more nuanced forms of causal reasoning, such as instrumental variable analysis, latent confounding, and policy evaluation. Additionally, we envision CausalPitfalls as a platform to guide training or fine-tuning strategies that aim to instill causal robustness in LLMs.

More detailed descriptions of our benchmark pitfall categories, challenges, and implementation details are included in the **Appendix** inside the supplementary material.

ACKNOWLEDGMENT

The work was supported in part by the National Science Foundation CAREER Program under grant number 2338506.

ETHICS STATEMENT

All authors have read and adhered to the ICLR Code of Ethics.

REPRODUCIBILITY STATEMENT

We have made every effort to ensure the reproducibility of our results. Detailed descriptions of datasets, preprocessing steps, and experimental settings are provided in the main text and appendix. The CausalPitfalls benchmark, including all datasets, questions, grading rubrics, and evaluation scripts, is publicly available at `https://github.com/dudududuu/CausalPitfalls`.

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
