## A    THE USE OF LARGE LANGUAGE MODELS (LLMS)

The authors acknowledge the use of ChatGPT to check for potential typographical and grammatical errors in the manuscript.

## B    BENCHMARKING KEY CHALLENGES IN RELIABLE CAUSAL INFERENCE FOR LLMS

### B.1    CONFOUNDING BIASES AND SPURIOUS ASSOCIATIONS

**Simpson's Paradox.**    Simpson's paradox is a causal pitfall where an observed association between two variables reverses or disappears when analyzed within subgroups defined by a confounding variable. This paradox typically arises when aggregated data conceal important subgroup-specific relationships, making an apparent correlation misleading or even contradictory.

*Motivation and Relevance:* Addressing Simpson's paradox is crucial because misinterpretations of aggregated relationships can lead to serious real-world consequence. LLMs are especially susceptible to this pitfall due to their tendency to overly rely on surface-level correlations and aggregated statistics, while neglecting underlying subgroup structures. Without specifically considering these subgroup, LLMs risk confidently delivering inaccurate causal conclusions.

*Data Generation:* We design five datasets to illustrate Simpson's paradox, and each dataset contains three binary variables representing a realistic medical scenario. Here's an example of data generation setting:

- Age Group (Confounder): Represents subgroup differences that might influence treatment assignment and outcomes (*Young* or *Old*).
- Drug Treatment (Treatment): Indicates whether a patient received a specific medication (*Drug given* or *No drug*).
- Recovery Status (Outcome): Represents patient recovery (*Recovered* or *Not recovered*).

The causal structure underlying the simulation is shown by the DAG below:

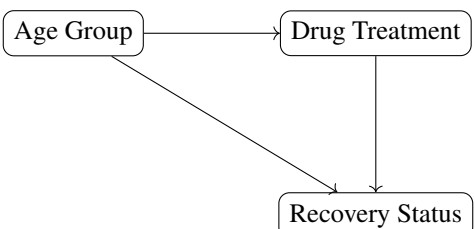

In this DAG, Age Group affects both the likelihood of receiving the drug treatment and the probability of recovery. Thus it confounds the relationship between treatment and recovery status. The generated datasets illustrate Simpson's paradox: when analyzed at an aggregate level, the drug appears beneficial, but within each age subgroup, the drug is actually harmful.

*Evaluation Rubric:* Models are evaluated based on their ability to recognize and correct for Simpson's paradox. Full credit (total score) is awarded if the LLM acknowledges the confounder, correctly identifies subgroup-specific relationships, and accurately states that the aggregate-level relationship is misleading. Partial credit is given if the model recognizes the discrepancy but fails to identify or correctly adjust for the confounder. No credit is awarded if the model relies only on aggregate-level associations without mentioning subgroup analysis or confounding.

**Berkson's Paradox**    Berkson's paradox is a causal pitfall arising from conditioning on a collider variable, thereby inducing a spurious association between two variables that are actually independent. Typically, this paradox appears when the analysis is restricted to a subgroup selected based on a variable influenced simultaneously by two independent factors, creating an artificial correlation.

*Motivation and Relevance:* Recognizing Berkson's paradox is essential, as it can lead to severely misguided conclusions. LLMs are particularly prone to this pitfall due to their reliance on readily accessible data summaries and correlations without carefully assessing selection mechanisms or collider structures. Without addressing collider-induced biases, LLMs may confidently report spurious relationships as meaningful causal insights.

*Data Generation:* We design five datasets to illustrate Berkson's paradox across realistic scenarios. Each dataset involves three binary variables, with one serving as a collider that induces selection bias. An example dataset represents a hospital-based scenario:

- Disease A: Indicates whether a patient has Disease A (*Present* or *Absent*).
- Disease B: Indicates whether a patient has Disease B (*Present* or *Absent*).
- Hospitalization (Collider): Represents whether the patient is hospitalized (*Yes* or *No*), influenced by both diseases.

The causal structure underlying Berkson's paradox is depicted by the DAG below:

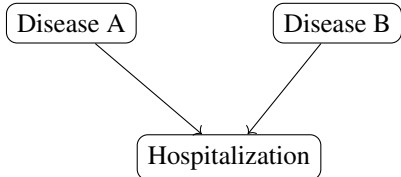

In this DAG, the variable *Hospitalization* is a collider, as it is influenced by Disease A and Disease B. When analyses are conditioned upon or restricted to hospitalized patients, it artificially creates an association between these two otherwise independent diseases. Thus, Berkson's paradox emerges through these generated datasets, underscoring the critical importance of avoiding inappropriate conditioning or properly correcting for selection biases.

*Evaluation Rubric:* Models are evaluated on their ability to recognize and appropriately correct for Berkson's paradox. Full credit is awarded if the LLM proposes the correct collider structure, states the independence assumption between the variables, proposes to use a valid bias-correction method (such as inverse probability weighting), and suggests no true association after correcting for collider bias. Partial credit is awarded if the model identifies collider bias but fails to state assumptions, proposes the correct correction approach, or suggests independence. No credit is given if the model neglects the collider issue altogether and draws conclusions based only on naïve associations without appropriate adjustment.

### B.2 Interventions and Experimental Reasoning

**Observational vs Experimental Reasoning** Observational reasoning refers to inferring causal relationships only from non-experimental, observational data, where the treatment is not randomly assigned. In contrast, experimental reasoning involves randomized controlled trials (RCTs), where treatment assignment is randomized, ensuring that confounding variables are evenly distributed across groups. The causal pitfall emerges when analysts interpret observational associations as causal effects without adequately adjusting for confounding variables, potentially resulting in incorrect causal conclusions.

*Motivation and Relevance:* Distinguishing between observational and experimental reasoning is crucial, as erroneous interpretations of observational data can significantly affect policy decisions, medical recommendations, and economic strategies. LLMs are particularly vulnerable to this pitfall due to their propensity to accept apparent observational correlations at face value, often neglecting confounders or the necessity of proper statistical adjustments. Without instruction or analytical rigor, LLMs may deliver misguided causal interpretations, amplifying the risk of poor decision-making in sensitive real-world applications.

*Data Generation:* We design five datasets to illustrate the differences between observational and experimental reasoning across realistic scenarios. Each dataset includes a binary treatment and a

binary outcome, along with several confounding variables. Here is an illustrative example scenario focusing on sleep quality:

- Supplement Use (Treatment): Indicates whether participants take a sleep-improving supplement (*Yes* or *No*).
- Sleep Quality (Outcome): Represents participants' quality of sleep (*High quality* or *Low quality*).
- True Confounders: Regular exercise habits (*Exercise regularly*) and older age (*Age old*), each affecting both supplement use and sleep quality.
- Irrelevant Variables (Non-confounders): Factors such as coffee drinking, watching TV late, and income, included to test the model's ability to discern relevant confounding from irrelevant variables.

The causal structure illustrating observational confounding is depicted by the DAG:

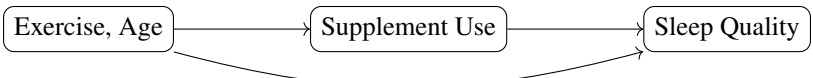

In this DAG, confounders (Exercise and Age) simultaneously affect both the treatment (Supplement Use) and the outcome (Sleep Quality), introducing bias into naïve observational analyses. Proper adjustment for these confounders is essential to approximate the true causal relationship.

*Evaluation Rubric:* Models are evaluated based on their capability to distinguish observational correlations from true causal effects. Full credit is awarded if the LLM correctly identifies true confounders, justifies their relevance, applies an appropriate statistical adjustment method (such as stratification or regression), and distinguishes the corrected causal conclusion from naïve observational results. Partial credit is awarded if the model recognizes the importance of confounding but fails to fully justify confounders or properly apply adjustments. No credit is given if the model relies only on observational correlations without addressing confounding or providing adjusted causal interpretations.

**Causal Effect Estimation** Causal effect estimation involves quantifying the precise magnitude of the effect that an intervention or treatment has on a particular outcome. Unlike qualitative assessments of causal relationships, effect estimation focuses on numerical measures, such as the average treatment effect (ATE). The primary challenge arises from properly accounting for confounding factors, distinguishing true causal effects from mere correlations or conditional probabilities.

*Motivation and Relevance:* Accurate estimation of causal effects is critical for informed decision-making in domains such as medicine, public policy, marketing, and technology. Misestimating these effects can lead to misguided interventions, inefficient resource allocation, or even unintended harm. LLMs often encounter difficulties with this task due to their inclination to rely on observational correlations without properly adjusting for confounders. Consequently, they risk confusing correlation or conditional probabilities with actual causal effects, potentially delivering flawed or misleading recommendations.

*Data Generation:* We design five datasets to evaluate the accuracy and robustness of causal effect estimation methods across realistic scenarios. Each dataset involves a defined binary treatment, binary outcome, multiple relevant confounders, and irrelevant features to test the ability of the model to distinguish between pertinent and non-pertinent information. An illustrative dataset focuses on patient recovery after secondary medical treatment:

- Secondary Treatment (Treatment): Indicates whether patients received a second-round medical treatment (*Received* or *Not received*).
- Recovery (Outcome): Reflects patient recovery status (*Recovered* or *Not recovered*).
- True Confounders: Factors such as patient age, initial illness severity, follow-up severity, and initial treatment, each influencing both the assignment of the secondary treatment and patient recovery.
- Irrelevant Features: Variables like socioeconomic status and random noise, intended to evaluate whether the model appropriately excludes irrelevant information.

The causal structure underlying these scenarios is represented by the DAG below:

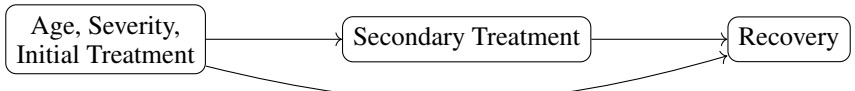

This DAG shows how confounding variables (Age, Severity, and Initial Treatment) simultaneously affect both the assignment of secondary treatment and the likelihood of recovery. Proper causal estimation requires statistical adjustments, such as inverse probability weighting (IPW), to isolate the true effect of the treatment from these confounding influences.

*Evaluation Rubric:* Models are evaluated based on their ability to accurately estimate causal effects and rigorously justify their methodological decisions. Full credit is awarded if the LLM defines the target causal estimand (such as ATE), correctly identifies and adjusts for true confounders, excludes irrelevant features, employs suitable causal inference methods (e.g., inverse probability weighting), provides numerical estimates within an acceptable tolerance range, and appropriately quantifies statistical uncertainty. Additionally, full-scoring responses perform at least one diagnostic check (e.g., balance or overlap assessments) and discuss methodological limitations. Partial credit is awarded if some, but not all, of these criteria are adequately addressed. No credit is given for relying only on unadjusted correlations without proper causal reasoning or methodological justification.

### B.3 COUNTERFACTUAL REASONING AND HYPOTHETICALS

**Counterfactual Outcome Prediction**   Counterfactual outcome prediction involves reasoning about hypothetical scenarios, specifically asking what would have happened if past events or treatments had occurred differently. This type of reasoning goes beyond mere correlation, requiring the careful consideration of alternative scenarios that contradict observed reality, while maintaining logical consistency with established causal relationships.

*Motivation and Relevance:* Counterfactual reasoning is essential for robust explanations, policy analysis, and informed decision-making across various disciplines, including medicine, economics, education, and environmental policy. Incorrect counterfactual predictions can lead to flawed policy recommendations and misguided interventions. LLMs often struggle with this task, as it requires moving beyond observed data toward hypothetical worlds that differ from reality. LLMs frequently either adhere too closely to observed facts or generate predictions that violate known causal structures, thereby compromising the reliability of their predictions.

*Data Generation:* We design five datasets for counterfactual outcome prediction tasks across diverse real-world contexts. Each dataset includes defined treatment, outcome, confounder, and downstream variables, enabling manipulation and assessment of counterfactual scenarios. One illustrative example is a clinical scenario involving drug dosage:

- Drug Dose (Treatment): Dosage levels administered to patients.
- Blood Concentration (Outcome): The resulting concentration level in patient blood following dosage.
- Baseline Health (Confounder): Initial health condition influencing both drug dosage and blood concentration.
- Therapeutic Score (Downstream Variable): A health outcome measure influenced by blood concentration but not directly adjustable in the counterfactual scenario.

The causal structure governing these scenarios is represented by the DAG:

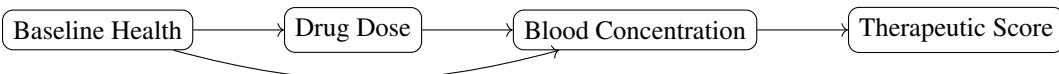

In this DAG, Baseline Health is a confounder affecting both the treatment (Drug Dose) and the outcome (Blood Concentration). The Therapeutic Score is a downstream variable, influenced only by the outcome and thus not to be controlled when evaluating counterfactuals. This challenge asks for

predictions under hypothetical alterations of the treatment variable, carefully ensuring consistency with the causal framework.

*Evaluation Rubric:* Models are evaluated based on their capability to accurately generate and reason about counterfactual outcomes. Full credit is awarded if the LLM identifies treatment, outcome, confounder, and downstream variables, proposes an appropriate causal DAG, acknowledges underlying assumptions (e.g., absence of hidden confounding), applies a valid method (such as regression adjustment) to estimate the counterfactual outcome, and provides numeric predictions that align closely with known counterfactual values. Partial credit is granted if the model correctly addresses some but not all aspects, such as identifying variables and providing accurate reasoning but failing to provide numerical accuracy or state assumptions. No credit is given if the model ignores causal structure, confounding, or essential assumptions, providing counterfactual predictions based only on observed correlations without causal reasoning.

**Causal Necessity and Sufficiency** Evaluating causal necessity and sufficiency involves determining whether a specific factor is required (necessary) or alone capable (sufficient) to produce an outcome. Necessity examines if the outcome would fail to occur in the absence of the cause, whereas sufficiency assesses if the presence of the cause alone invariably leads to the outcome. This analysis is particularly crucial in scenarios with multiple, potentially redundant or overlapping causal factors.

*Motivation and Relevance:* Correctly assessing causal necessity and sufficiency is fundamental for rigorous explanation, accountability, and policy decisions. Misclassification can lead to incorrect attribution, misguided interventions, or failure to identify effective alternative measures. LLMs commonly encounter difficulties with this sophisticated causal reasoning; they often erroneously assume necessity in the presence of alternative sufficient causes or incorrectly attribute sufficiency without thorough evaluation. Expert causal reasoning involves considering counterfactual scenarios — imagining the outcome under conditions where a causal factor is removed or is the sole influencing factor.

*Data Generation:* We simulate five datasets designed to test the understanding of causal necessity and sufficiency across diverse applied contexts. Each scenario includes a focal causal factor ($X$), outcome variable ($Y$), and additional contextual factors. An illustrative example involves material stress in engineering structures:

- Material Stress (Focal Cause): Evaluates whether elevated material stress is necessary or sufficient for structural failure.
- Failure Probability (Outcome): Indicates whether structural failure occurs beyond a specific threshold.
- Contextual Factors: Load pressure and vibration frequency, serving as potential alternative or complementary causal factors influencing failure.

The causal structure underlying necessity and sufficiency is illustrated by the DAG:

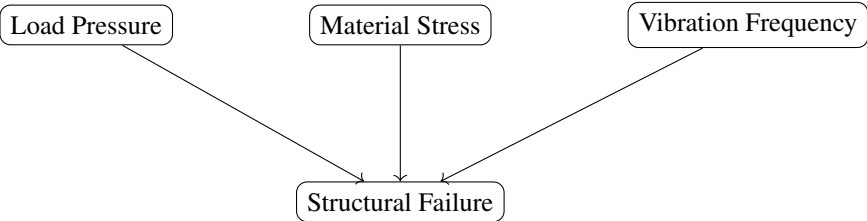

In this DAG, the focal causal factor (Material Stress) is evaluated alongside alternative or complementary causes (Load Pressure, Vibration Frequency) to determine if it is necessary or sufficient for structural failure. Proper analysis requires reasoning about interventions and counterfactual scenarios, comparing outcomes under varying causal conditions.

*Evaluation Rubric:* Models are evaluated based on their capability to rigorously define and correctly classify causal necessity and sufficiency. Full credit is awarded if the LLM defines global necessity and sufficiency using accurate counterfactual reasoning, mentions critical thresholds, thoroughly analyzes outcomes under varied interventions, describes potential outcomes , and correctly classifies

the causal factor based on the provided scenarios. Partial credit is awarded for responses that partially fulfill these criteria—such as accurately defining concepts and analyzing some interventions but not comprehensively covering all required elements. No credit is given if the model fails to apply proper counterfactual logic, ignores alternative causes, or incorrectly classifies necessity and sufficiency without justification.

## B.4 MEDIATION AND INDIRECT CAUSAL EFFECTS

Mediation analysis focuses on understanding how a treatment influences an outcome through intermediate variables known as mediators. Identifying these indirect pathways accurately is essential, as interventions typically propagate through multiple, intertwined channels. However, precise estimation of mediated effects raises methodological challenges.

**Mediator–Outcome Confounding**   Mediator–outcome confounding arises when a third variable influences both the mediator ($M$) and the outcome ($Y$), biasing the estimated indirect (mediation) and direct effects of a treatment ($T$). In the canonical structure $T \to M \to Y$, a confounder ($C$) that affects both $M$ and $Y$ distorts mediation estimates unless it is properly controlled. The task therefore requires distinguishing true mediation pathways from correlations induced by common causes of the mediator and the outcome.

*Motivation and Relevance:* Failure to address mediator–outcome confounding can lead to severely biased causal interpretations and misguided policies. For instance, when evaluating whether physical activity mediates the effect of an educational program on cognitive performance, socioeconomic status may independently influence both activity levels and test scores. LLMs frequently overlook this bias: they may attribute the entire treatment effect to the mediating variable or misestimate indirect and direct effects because they ignore hidden pathways. Correct reasoning demands recognising potential confounders of $M$ and $Y$ and applying specialised methods, such as sequential g-estimation, inverse-probability–weighted mediation analysis, or parametric g-formulae, to obtain unbiased mediation effects.

*Data Generation:* We construct five datasets that embed mediator–outcome confounding in realistic settings. A representative scenario examines an educational intervention:

- Treatment ($T$): Receipt of a study-skills intervention (*Yes / No*).
- Mediator ($M$): Weekly physical-activity hours.
- Outcome ($Y$): Post-intervention cognitive test score.
- Confounder ($C$): Socioeconomic status (SES), influencing both activity and test performance.

The causal structure is illustrated by the DAG below:

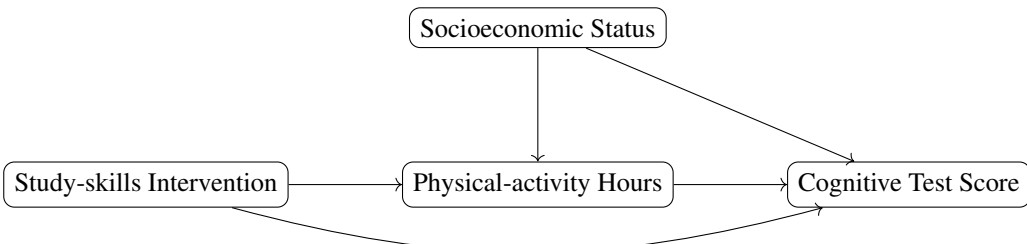

*Evaluation Rubric:* Models are evaluated on their ability to detect and correct for mediator–outcome confounding. Full credit is awarded when the response identifies the confounder that influences both the mediator and the outcome, employs and justifies a valid adjustment method (for example, sequential $g$-estimation or inverse-probability weighting), provides numerical estimates of the natural direct and indirect effects within the specified tolerance, reports and interprets a measure of statistical uncertainty (such as a confidence interval or standard error). Partial credit is given when only some of these elements are addressed. For instance, if the confounder is identified and a method applied but uncertainty quantification or a limitations discussion is omitted. No credit is awarded if

mediator–outcome confounding is ignored, only unadjusted mediation estimates are presented, or methodological choices are left unjustified.

**Sequential Mediators**    Sequential mediation occurs when an intervention exerts its effect on an outcome through a chain of two or more intermediate variables. The task requires disentangling each step in the mediator sequence: identifying how the treatment first influences one mediator, how that mediator then affects the next, and so on until the final outcome.

*Motivation and Relevance:* Many real-world interventions operate via multiple stages. For example, an educational program that first enhances coping strategies, which in turn builds social support, ultimately improving mental health. Ignoring the sequential nature of these mediators can misattribute effects to the wrong pathway and lead to suboptimal or even counterproductive interventions. LLMs are especially challenged by this scenario because they must recognize the ordered dependencies among mediators and apply methods (such as path-specific effects analysis or structural equation models) rather than treating mediators as independent channels.

*Data Generation:* We simulate five datasets reflecting a realistic educational intervention scenario:

- Educational Intervention (Treatment): Whether students receive a skills-training program.
- Coping Strategy Improvement (Mediator 1): Increase in students' coping skills following the intervention.
- Social Support Enhancement (Mediator 2): Growth in support networks resulting from improved coping.
- Mental Health Score (Outcome): Final assessment of students' psychological well-being.
- Confounders: Socioeconomic status, school quality, and baseline cognitive score (each affecting treatment assignment and mediators).

The underlying causal structure is depicted by the DAG:

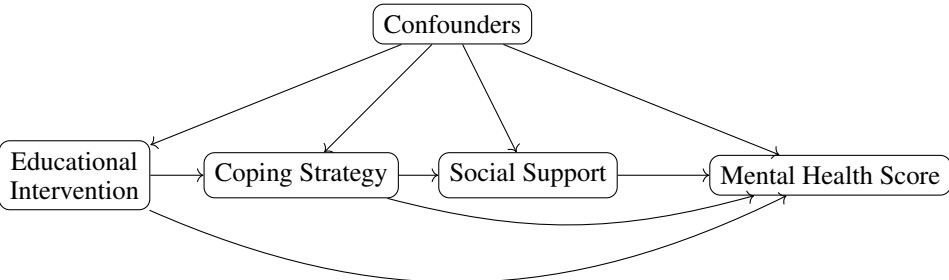

*Evaluation Rubric:* Models are evaluated on their ability to recover the sequential mediation structure and produce unbiased path-specific estimates. Full credit is awarded if the response correctly identifies all true confounders and both mediators in their proper order, excludes irrelevant variables, selects and justifies an appropriate sequential mediation method (for example, path-specific effects analysis or structural equation modeling), contrasts adjusted indirect effects with naïve estimates, provides numerical effect estimates within the specified tolerance, reports and interprets statistical uncertainty, and acknowledges observational limitations. Partial credit is given when only some of these elements are addressed (for instance, correctly identifying mediators but omitting uncertainty quantification). No credit is given if the model ignores the sequential pathway, applies only naïve mediation analysis, or fails to justify methodological choices.

**Treatment–Mediator Interaction Effects**    Treatment–mediator interaction arises when the magnitude of a treatment's effect on an outcome varies with the level of a mediator. In other words, the indirect pathway through the mediator modifies the direct effect of the treatment. Ignoring these interactions can mask important synergies or antagonisms, leading to incorrect estimates of direct and indirect effects.

*Motivation and Relevance:* Accurately modeling treatment–mediator interactions is vital because policies or interventions may only be effective under specific mediator conditions. For example, an

economic stimulus might only boost employment significantly when consumer confidence is already high. LLMs often default to additive causal models and overlook such interaction terms, risking misleading conclusions and suboptimal policy or clinical recommendations.

*Data Generation:* We design five datasets to illustrate treatment–mediator interaction effects in an economic context.

- Stimulus Policy (Treatment): Indicates whether a region implements an economic stimulus (*Yes/No*).
- Consumer Confidence (Mediator): Public confidence level (*High/Low*), which may modify the policy's effect.
- Employment Rate (Outcome): Regional employment outcome (*Increased/Not increased*).
- Confounders: Market volatility and baseline economic growth, each affecting treatment assignment, consumer confidence, and employment.
- Irrelevant Variable: An unrelated indicator, included to test the model's ability to exclude non-pertinent features.
- Interaction Term: The product of Stimulus Policy and Consumer Confidence, included in the outcome model to capture effect modification.

The underlying causal structure is depicted by the DAG below:

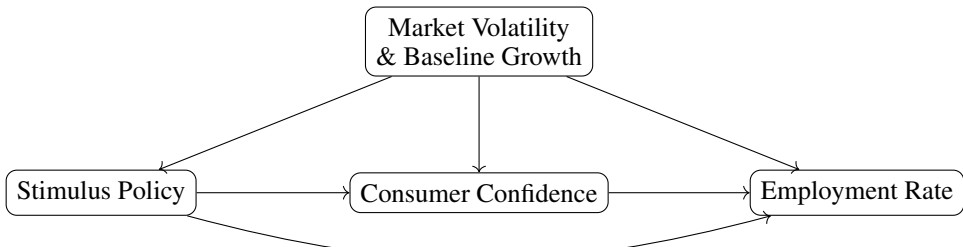

*Evaluation Rubric:* Models are evaluated on their ability to identify true confounders and the mediator, articulate how the treatment–mediator interaction modifies the causal effect, justify exclusion of irrelevant variables, select and justify an interaction-aware estimation method (such as regression with an interaction term), compare adjusted estimates to naïve associations, and provide numerical results with uncertainty measures. Full credit requires all these elements; partial credit is given when only some are addressed; no credit is awarded if interactions are ignored or methodological justifications are missing.

### B.5    CAUSAL DISCOVERY AND STRUCTURE LEARNING

Causal discovery and structure learning involve identifying unknown causal relationships and constructing comprehensive causal graphs from observational or experimental data. This type of analysis is distinct from tasks that estimate causal effects or perform mediation analysis. The primary goal here is to discover if causal links exist, determine their direction, and outline the overall causal structure from scratch. The process includes analyzing patterns of statistical dependence and conditional independence among variables. This can be particularly challenging for LLMs because they typically rely on correlations rather than deeper causal insights.

**Cause–Effect Direction Inference**    Determining the correct causal direction between two correlated variables, deciding whether $X_1 \rightarrow X_2$ or $X_2 \rightarrow X_1$, is fundamental to causal discovery. This task requires more than observing associations; it demands identifying which variable truly generates the other, often by leveraging auxiliary information beyond the raw correlation.

*Motivation and Relevance:* Inferring the wrong direction can lead to ineffective or harmful interventions, as policies and predictions depend critically on understanding which variable to manipulate. LLMs typically struggle here because they rely on surface-level correlations or heuristics. In practice,

experts use additional clues (such as instrumental variables, temporal ordering, or distributional asymmetries) to distinguish cause from effect.

*Data Generation:* We design five datasets in which two exogenous instruments $(Z_1, Z_2)$ each influence exactly one of two target variables $(X_1, X_2)$, and a single true causal edge runs between $X_1$ and $X_2$. An illustrative dataset is generated as follows:

- $Z_1, Z_2$ (*Instruments*): Independent variables that affect $X_1$ or $X_2$ respectively, but not each other.
- $X_1, X_2$ (*Targets*): Two correlated variables linked by a true causal arrow (e.g. $X_1 \rightarrow X_2$).
- Noise terms: Independent additive errors for each variable.

The causal structure for this illustrative dataset is depicted below:

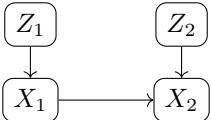

Here, each instrument $Z_i$ is designed to have a nonzero effect only on the corresponding $X_i$, and the single arrow $X_1 \rightarrow X_2$ represents the ground-truth causal direction.

*Evaluation Rubric:* Models are evaluated on their ability to assign instruments correctly, carry out the two required regressions of $X_1$ and $X_2$ on $\{Z_1, Z_2\}$, report $p$-values and identify which instrument is significant in only one regression, infer the causal arrow accordingly, and match that inference to the known truth. Full credit is awarded when all these elements are present; partial credit when some are addressed; and no credit if the solution relies only on correlations or omits any required component.

**Handling Uncertainty in Causal Structures**   Handling uncertainty in causal structure inference requires acknowledging that, for a given set of observed conditional independencies, multiple Directed Acyclic Graphs (DAGs) may be equally compatible with the data. This phenomenon, known as Markov equivalence, means that one cannot definitively pinpoint a single causal graph without additional assumptions or interventions.

*Motivation and Relevance:* Properly communicating uncertainty over plausible causal models is essential in fields such as epidemiology, economics, and social science, where decisions hinge on structural conclusions. Overconfident or overly definitive causal claims can mislead policy-makers or clinicians. While human experts often report equivalence classes or express confidence intervals over edges, LLMs tend to present a single "best" graph, failing to convey ambiguity or the need for further evidence.

*Data Generation:* We sample five SEM datasets, and in each case, the true data-generating DAG is a simple forward chain $X_1 \rightarrow X_2 \rightarrow X_3$, and independent gaussian noise terms are added with no hidden confounding. The model is chosen that the forward chain and its two Markov-equivalent counterparts are indistinguishable by purely observational tests.

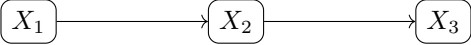

*Evaluation Rubric:* Models are evaluated on whether they perform necessary marginal and conditional independence tests, list all Markov-equivalent DAGs (including the forward chain, its reverse-fork, and the reverse chain) without proposing any unsupported structures, and refer to Markov equivalence or equivalence-class ambiguity rather than presenting a single graph with unwarranted certainty. Full credit requires correct recognition and expression of uncertainty among all valid graphs; partial credit is given for acknowledging some but not all aspects of the equivalence class or uncertainty; no credit is awarded if the model reports a single graph as definitive or omits the equivalence consideration altogether.

### B.6   CAUSAL GENERALIZATION AND EXTERNAL VALIDITY

Causal generalization and external validity address the critical issue of whether a causal relationship identified in one context remains valid when applied to another context. Unlike internal validity,

which confirms that observed effects are truly due to the intervention within a specific setting, external validity assesses the transferability of causal claims across varying conditions, populations, or environments. LLMs often assume learned causal relationships universally apply, failing to detect subtle contextual differences that could invalidate such assumptions. Robust causal reasoning thus demands recognizing the limits and scope of causal knowledge.

**Population Shift and Transferability.** Population shift arises when causal effects estimated in one group (e.g. Population A) do not generalize to another group (Population B) due to differences in baseline characteristics or effect modification. Transferability requires recognising when and how causal conclusions must be adjusted before being applied across populations.

*Motivation and Relevance:* Failing to account for population shift can lead to ineffective or harmful interventions when policies validated in one demographic are indiscriminately exported to another. For example, an educational program that boosts test scores in urban schools may underperform or backfire in rural settings with different socioeconomic profiles. LLMs often extrapolate causal claims without acknowledging population nuances; robust methods are needed to ensure safe and effective transfer.

*Data Generation:* We design five datasets comparing two populations ($A$ vs. $B$). Each record includes:

- Population: Indicator ($A$ or $B$), determining which structural parameters apply.
- SocioEconomic Status: A baseline covariate influencing both program uptake and outcomes.
- Program Intensity: The "dose" of an educational intervention.
- Test Score: The measured outcome.

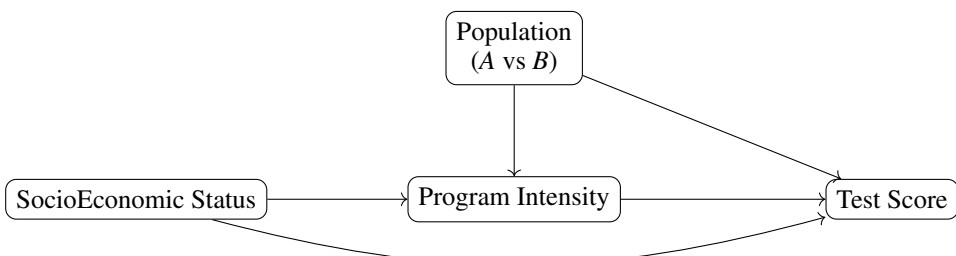

*Evaluation Rubric:* Models are evaluated on their ability to distinguish Population $A$ and $B$, report the coefficient of Program Intensity on Test Score (with statistical significance or uncertainty) for each population, compare the magnitudes of these effects across populations, and summarize the implications for transferability. Full credit is awarded when all these elements are present and correctly interpreted; partial credit when only some are addressed; and no credit if the model ignores population distinctions or fails to compare and contextualize effect estimates across groups.

**Temporal Stability of Causal Effects** Temporal stability examines whether a causal effect remains constant across different time periods or whether it shifts due to changing conditions. The task requires detecting and quantifying any differences in the treatment's impact on the outcome between an initial period and a later period.

*Motivation and Relevance:* Evaluating temporal stability is vital for reliable forecasting and adaptive policy-making. An intervention that drives strong effects early on may weaken, or even reverse, its impact as underlying dynamics evolve. LLMs often assume static relationships, risking misleading guidance when effects drift. Experts counter this by stratifying analyses by time period, including interaction terms with time, or using time-varying coefficient models to capture and adjust for temporal change.

*Data Generation:* We simulate five datasets illustrating temporal drift in a user-engagement scenario. Each record includes:

- Time Period: Indicator of Period 1 or Period 2.
- User Invites (Treatment): Number of invitations sent to potential users.

- User Signups (Outcome): Number of users who sign up.

The causal structure is depicted by the DAG below:

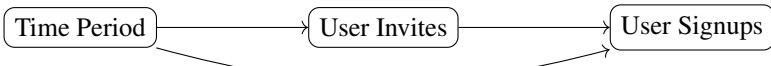

*Evaluation Rubric:* Models are evaluated on their ability to distinguish Period 1 from Period 2, fit separate analyses of signups on invites for each period, report the estimated effect sizes within the specified tolerance, discuss statistical significance, interpret whether the effect has strengthened or weakened over time, and conclude appropriately about temporal variation. Full credit is awarded when all these elements are present; partial credit if only some aspects are addressed; and no credit if temporal drift is ignored or treated as time-invariant.

**Contextual Interaction and Moderation Effects**    Contextual interaction, or moderation, occurs when the effect of a treatment on an outcome depends on the level of another variable (the moderator). The task is to identify and quantify how this moderator alters the treatment's impact, rather than assuming a uniform effect across all contexts.

*Motivation and Relevance:* Accounting for moderation is critical because interventions may only be effective under particular conditions. For example, a pain medication might relieve symptoms in mild arthritis but be less effective or even counterproductive in severe cases. LLMs often overlook such nuances, treating effects as homogeneous and risking flawed recommendations. Experts model interaction terms or conduct stratified analyses to capture these context-dependent effects.

*Data Generation:* We design five datasets illustrating dosage–severity interactions in clinical and biological settings.

- Condition Severity: A continuous or ordinal measure of baseline severity.
- Treatment Dosage: Dosage level administered.
- Symptom Reduction: The observed decrease in symptoms.
- Interaction Term: The product of Condition Severity and Treatment Dosage is included in the outcome model to generate a moderation effect.

The causal structure is represented by the DAG below:

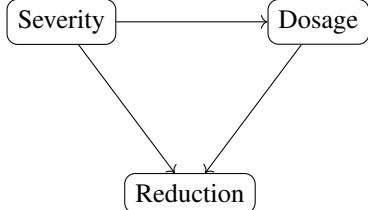

*Evaluation Rubric:* Models are evaluated on their ability to report the main effect of dosage, the main effect of severity, and the interaction coefficient; to interpret how severity levels moderate the dosage effect; to perform and mention any diagnostic checks (e.g. residual analysis); and to summarize the implications for tailored dosing. Full credit is awarded when all elements are addressed with appropriate numerical or qualitative detail; partial credit when only some are; and no credit if moderation is ignored or methodological justifications are omitted.

**Domain Shift and Transportability of Causal Knowledge**    Domain shift arises when causal conclusions drawn in one population fail to generalize to another due to differences in baseline characteristics or effect modifiers. Transportability concerns identifying these differences and adjusting causal inferences so that interventions remain valid across domains.

*Motivation and Relevance:* Ignoring domain shift can lead to interventions that backfire or underperform when moved to new settings. For example, a drug shown to reduce blood pressure in

middle-aged adults may be less effective in older patients with higher frailty. LLMs often extrapolate causally without accounting for such nuances, whereas experts employ statistical techniques to pinpoint and correct for population differences, ensuring safe and effective transfer of causal knowledge.

*Data Generation:* We design five datasets contrasting two age-defined cohorts (Group 1: ages 30–50; Group 2: ages 50–70).

- Age: Numeric age to determine the membership of the cohort.
- Frailty: A health indicator positively correlated with age.
- Drug Dose (Treatment): Dose of drug taken.
- Blood Pressure Reduction (Outcome): Measured drop in systolic blood pressure.

This design forces models to grapple with shifting treatment effects under different demographic distributions.

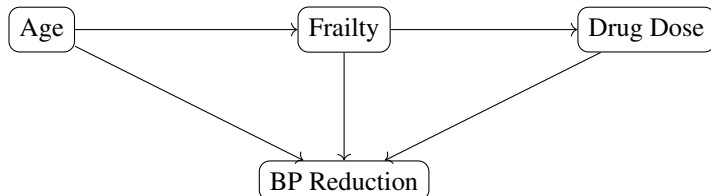

*Evaluation Rubric:* Models are evaluated on their ability to propose splitting the data into the correct age strata, specify and justify a linear model of blood pressure reduction on drug dose and frailty within each stratum, condition the estimate on the individual's age scenario (e.g. age = 65), recognize frailty as a necessary input for precise effect estimation, explain how frailty confounds the dose–response in the older cohort, and request any additional information needed for a complete answer. Full credit is awarded when all elements are addressed; partial credit when only some are present; and no credit if domain differences are ignored or methodological steps are omitted.

## C    DETAILS OF TWO PROTOCOLS

We evaluated models on causal inference tasks using two different prompting approaches: **Direct Prompting** and **Code Assisted Prompting**. Each method tests a distinct aspect of the model's abilities.

### C.1    PROTOCOL 1: DIRECT PROMPTING

In the **Direct Prompting** protocol, we test the model's ability to analyze causal questions directly from raw data, without using computational code.

**Workflow for Direct Prompting**

1. **Model Prompting:** Present the question alongside the raw data to the model.
2. **Output Collection:** Record the model's response for evaluation.

**Example:**

> *"Question: Evaluate whether {TREATMENT} causally affects {OUTCOME}.*
>
> *Data: {data}*
>
> *Provide your analysis without using code."*

This method focuses on the model's ability to reason intuitively and draw conclusions directly from the provided data.

### C.2    PROTOCOL 2: CODE-ASSISTED PROMPTING

The **Code-Assisted Prompting** protocol tests whether the model can identify causal issues, generate relevant Python code to analyze the data, and interpret the numerical outcomes to resolve the causal question.

**Workflow for Code Assisted Prompting**

1. **Code Generation:** Provide the model with the causal question, dataset location, column names, and a small data sample (10 rows). Request Python code for analysis.
2. **Code Execution:** Extract and run the generated Python code to obtain numerical results.
3. **Result Interpretation:** Show the model the code it generated and its numerical results, and ask it to interpret these results in context.
4. **Output Collection:** Record the model's interpretation and analysis for evaluation.

**Example (Code Generation Prompt):**

> *"Question: Evaluate whether {TREATMENT} causally affects {OUTCOME}.*
>
> *Dataset location:* `/path/to/dataset.csv`
>
> *Columns:* {`col1, col2, col3, ...`}
>
> *Data sample (first 10 rows):* {data_sample}
>
> *Provide Python code to perform this analysis."*

**Example (Results Interpretation Prompt):**

> "Question: Evaluate whether {TREATMENT} causally affects {OUTCOME}.
>
> Code:
>
> {Python code generated by the model}
>
> Results obtained:
>
> {Numerical results from code execution}
>
> Based on these results, provide your analysis and answer the causal question."

Together, the two protocols provide complementary insights, evaluating both the intuitive reasoning and computational planning of models in causal inference tasks.

# D    CAUSAL PITFALL EXAMPLES

This appendix provides detailed descriptions of two illustrative examples as introduced in Section 3.

## D.1    BRANDING BIAS: ADVERSARIAL SENSITIVITY TO BRANDING AND SEMANTIC MANIPULATION

To highlight the susceptibility of LLMs to semantic manipulations, we designed an adversarial scenario testing whether beverage branding influences LLM-based causal conclusions about health impacts. The underlying causal structure was fixed, ensuring beverage consumption directly affected health outcomes positively or negatively, while lifestyle and health awareness independently affected both beverage consumption and health. Importantly, the beverage brand names ("HealthPlus" or "UltraSugar") themselves had no actual causal effect (see Figure 3 in the main paper).

**Dataset Construction.**    We generated two datasets to represent four distinct scenarios, each combining a brand name ("HealthPlus", "UltraSugar") with a true health effect (beneficial or harmful):

1. Brand "HealthPlus", truly beneficial effect.
2. Brand "HealthPlus", truly harmful effect.
3. Brand "UltraSugar", truly harmful effect.
4. Brand "UltraSugar", truly beneficial effect.

Each dataset included 200 samples with variables: Consumption, Outcome (health impact), Health Awareness, and Lifestyle.

**LLM Performance and Observations.**    LLMs were asked to assess if each beverage ("HealthPlus" or "UltraSugar") was beneficial or harmful based purely on the given data. Table 3 in the main paper summarizes the LLM conclusions across scenarios.

## D.2    SPURIOUS CAUSAL INFERENCE FROM RANDOM PATTERNS.

We analyze the dataset `research_funding_rates` from the `dslabs` R package to illustrate how a rigorous causal analysis should be conducted.

We first construct a dataset containing the number of applications, awards, and success rates for each gender. Disciplines are then re-ordered by their overall success rate from the original dataset. To examine whether the data exhibit Simpson's paradox, we plot success rates by discipline (ordered by overall success), using color to indicate gender and point size to reflect the number of applications.

As shown in Figure 5, there is no clear confounder driving the observed patterns. Nonetheless, some fields appear to favor men, while others favor women. Notably, the two disciplines with the largest differences favoring men are also those with the largest number of applications.

This raises the question: could some selection committees be biased while others are not? To investigate, we compute the log-odds ratio divided by its standard error for each discipline and examine their distribution using a Q–Q plot against the standard normal distribution.

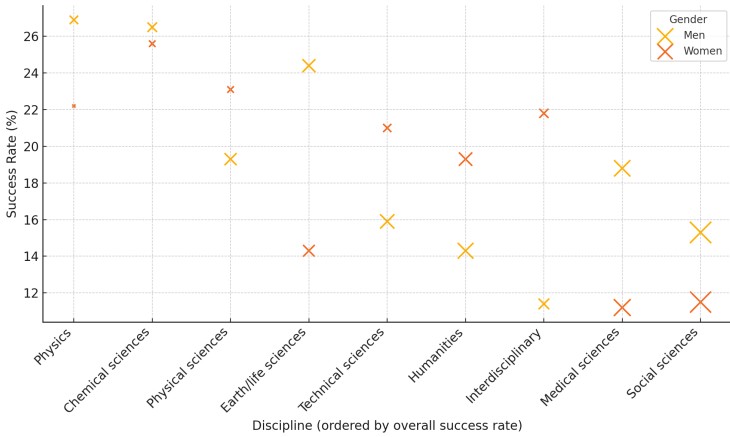

Figure 5: Success rates across disciplines for men and women.

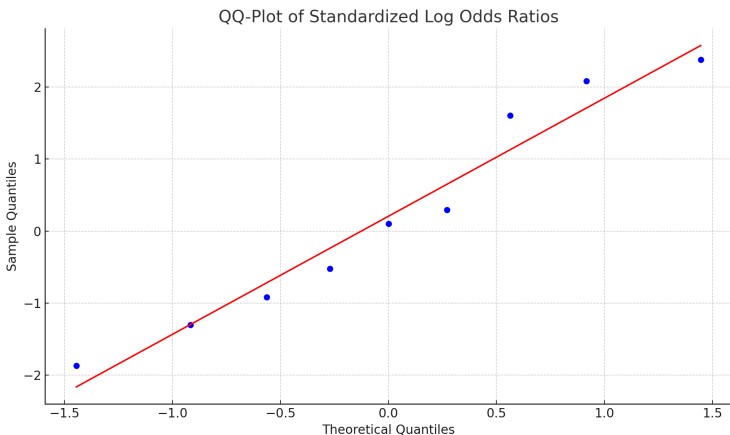

Figure 6: QQ-plot of standardized log odds ratios across disciplines.

Figure 6 shows that the points lie close to the reference line, indicating that the observed effect sizes across disciplines are consistent with random variation under the null hypothesis of no systematic bias. The single "significant" result in Medical Sciences, which emerges under a chi-square test, is plausibly attributable to chance when conducting multiple comparisons, rather than reflecting genuine committee-level bias.

# E    HUMAN-LLM GRADING ALIGNMENT

To assess the reliability of our GPT-4o-based automated evaluation approach, we conducted a validation involving expert human graders. This section elaborates on our detailed validation procedure.

**Sampling Procedure.**    We randomly selected 150 LLM-generated responses from our complete evaluation dataset, employing **stratified** random sampling to ensure balanced representation across:

1. First, we divided responses into the six causal inference categories, selecting an equal number (25) from each category.

2. Within each category, we evenly sampled across the five difficulty levels (very easy, easy, medium, hard, very hard), selecting exactly 5 responses per difficulty level.

3. For each category-difficulty combination, we randomly selected responses from the evaluated LLMs, ensuring proportional representation of all models' outputs.

This sampling approach ensured the validation set accurately represented the complexity, diversity, and balanced coverage of our entire evaluation dataset.

**Human Grading Procedure.**  Each sampled response was independently evaluated by three PhD students majoring in statistics. Prior to grading, the evaluators received detailed instructions on the grading rubrics to ensure consistency in scoring. Each grader assessed responses independently based on these rubrics, after which we aggregated their individual scores by taking the arithmetic mean. The average *Gap* (absolute scoring differences) between human-expert and GPT-4o automated scores across causal inference categories, are summarized in Table 6.

Table 6: Average *Gap* (absolute scoring differences) between human graders and GPT-4o across causal inference categories (25 samples per category).

| Causal Pitfall Category | Average Gap |
|---|---|
| Confounding biases and spurious associations | 0.08 |
| Interventions and experimental reasoning | 0.05 |
| Counterfactual reasoning and hypotheticals | 0.15 |
| Mediation and indirect causal effects | 0.17 |
| Causal discovery and structure learning | 0.15 |
| Causal generalization and external validity | 0.05 |
| **Overall (all categories)** | **0.11** |

# F  ADDITIONAL EXPERIMENTAL SETUP

Our experiments included the following ten large language models: Llama3.1-8b, Llama3.1-70b, Mistral-7b, Mixtral-8x22b, Claude-3.5-sonnet, Gemma2-9b, Gemini-2.0-flash, Deepseek-chat, GPT-4.1, and GPT-o4-mini.

All experiments and analyses were conducted using a single NVIDIA A100 GPU. API interactions were handled using the official SDKs provided by each model vendor (OpenAI, Anthropic, Google, Deepseek).

Hyperparameters were consistently set across models to ensure fair comparisons. The maximum token length was standardized at 1000 tokens for all models. GPT o4-mini was configured using the parameter `reasoning_effort="medium"`.

# G  ADDITIONAL EXPERIMENTAL RESULTS

Tables 7 and 8 provide comprehensive evaluations of LLM reliability at the level of individual causal inference challenges, under both direct and code-assisted prompting protocols.

# H  CODE-ASSISTED PROMPTING WITH DEBUGGING

To further investigate whether the performance gap under code-assisted prompting arises from unreliable code generation, we introduce a variant: **Code-Assisted Prompting with Debugging**. This protocol extends Protocol 2 by allowing the model one opportunity to correct its code when execution fails. Specifically, when the generated code raises an error, we present the error message to the model and request a fixed version and re-execute the code.

Table 7: Direct Prompting: normalized scores (%) of LLMs across all individual causal inference challenges.

| LLM | CE | CD | CI | CF | DS | MC | NS | OE | PS | SB | SM | SP | SU | TS | TM |
|-----|-----|-----|-----|-----|-----|-----|-----|-----|-----|-----|-----|-----|-----|-----|-----|
| Gemma2-9b | 38.50 | 2.40 | 2.00 | 15.43 | 11.33 | 25.00 | 2.21 | 22.94 | 0.00 | 4.00 | 19.50 | 24.00 | 16.57 | 2.86 | 5.00 |
| Llama3.1-8b | 40.00 | 4.00 | 4.00 | 21.71 | 15.33 | 38.50 | 1.54 | 29.56 | 0.00 | 7.20 | 29.00 | 29.71 | 11.43 | 6.86 | 20.50 |
| Llama3.1-70b | 37.00 | 7.20 | 7.33 | 20.00 | 16.67 | 43.00 | 3.87 | 34.83 | 5.60 | 7.20 | 36.50 | 28.00 | 15.43 | 15.43 | 21.50 |
| Mistral-7b | 37.00 | 4.80 | 2.00 | 10.86 | 21.33 | 18.50 | 2.04 | 22.67 | 0.80 | 7.20 | 25.00 | 27.43 | 12.00 | 0.57 | 14.00 |
| Mixtral-8x22b | 39.50 | 6.40 | 5.33 | 16.00 | 18.00 | 35.00 | 2.64 | 24.67 | 4.80 | 8.80 | 33.00 | 24.00 | 12.00 | 10.86 | 14.50 |
| Claude-3.5-sonnet | 51.88 | 15.20 | 13.33 | 17.71 | 18.06 | 59.00 | 9.73 | 43.33 | 25.60 | 7.20 | 40.50 | 30.29 | 22.36 | 22.29 | 22.00 |
| Gemini-2.0-flash | 43.00 | 9.60 | 10.67 | 20.00 | 25.69 | 60.50 | 9.87 | 30.76 | 11.20 | 6.40 | 48.50 | 33.71 | 20.41 | 12.57 | 31.00 |
| Deepseek-chat | 58.00 | 20.80 | 19.33 | 24.00 | 25.33 | 80.50 | 5.80 | 46.83 | 40.00 | 15.20 | 48.00 | 36.57 | 20.88 | 30.29 | 33.00 |
| GPT-4.1 | 36.72 | 18.40 | 16.67 | 10.29 | 22.67 | 65.79 | 6.59 | 31.56 | 38.40 | 8.80 | 61.31 | 25.71 | 14.29 | 18.29 | 37.00 |
| GPT-o4-mini | 50.62 | 40.80 | 32.00 | 25.14 | 28.67 | 83.00 | 16.51 | 39.79 | 54.40 | 36.00 | 52.50 | 46.86 | 33.14 | 62.86 | 37.50 |
| **Average** | 43.22 | 12.96 | 11.27 | 18.11 | 20.31 | 50.88 | 6.08 | 32.69 | 18.08 | 10.80 | 39.38 | 30.63 | 17.85 | 18.29 | 23.60 |

CE: Causal effect estimation; CD: Cause-effect direction; CI: Contextual interaction; CF: Counterfactual prediction; DS: Domain shift. MC: Mediator-outcome confounding; NS: Necessity and sufficiency; OE: Observational vs experimental; PS: Population shift; SB: Selection bias. SM: Sequential mediators; SP: Simpson's paradox; SU: Structure uncertainty; TS: Temporal stability; TM: Treatment-mediator interaction.

Table 8: Code-Assisted Prompting: normalized scores (%) of LLMs across all individual causal inference challenges.

| LLM | CE | CD | CI | CF | DS | MC | NS | OE | PS | SB | SM | SP | SU | TS | TM |
|-----|-----|-----|-----|-----|-----|-----|-----|-----|-----|-----|-----|-----|-----|-----|-----|
| Gemma2-9b | 21.50 | 6.40 | 32.67 | 9.71 | 6.00 | 18.12 | 1.73 | 15.50 | 1.60 | 7.20 | 32.00 | 8.57 | 6.86 | 33.71 | 18.00 |
| Llama3.1-8b | 19.44 | 6.96 | 15.97 | 12.57 | 21.53 | 17.50 | 1.69 | 12.72 | 7.20 | 5.00 | 21.59 | 22.32 | 5.44 | 24.57 | 15.83 |
| Llama3.1-70b | 23.81 | 16.80 | 33.33 | 16.07 | 23.33 | 37.50 | 3.80 | 22.00 | 40.00 | 12.80 | 19.50 | 34.29 | 11.69 | 4.00 | 18.50 |
| Mistral-7b | 15.91 | 3.20 | 20.29 | 2.29 | 14.67 | 12.00 | 1.90 | 10.51 | 1.80 | 0.80 | 16.48 | 9.09 | 9.94 | 2.29 | 5.50 |
| Mixtral-8x22b | 29.17 | 18.18 | 30.67 | 9.94 | 20.29 | 33.70 | 2.90 | 31.67 | 16.80 | 11.67 | 26.70 | 33.33 | 9.77 | 29.14 | 18.75 |
| Claude-3.5-sonnet | 53.95 | 13.64 | 24.64 | 17.14 | 23.61 | 48.44 | 9.57 | 40.72 | 20.80 | 18.40 | 48.91 | 47.43 | 19.33 | 36.57 | 28.12 |
| Gemini-2.0-flash | 50.00 | 14.40 | 54.67 | 21.14 | 25.69 | 53.00 | 10.52 | 34.17 | 24.80 | 18.40 | 40.50 | 56.00 | 19.39 | 46.29 | 33.00 |
| Deepseek-chat | 48.96 | 28.00 | 46.00 | 20.00 | 26.00 | 61.00 | 5.37 | 48.44 | 56.80 | 18.40 | 47.50 | 58.86 | 20.78 | 53.71 | 32.29 |
| GPT-4.1 | 37.50 | 26.40 | 51.96 | 21.14 | 28.00 | 60.16 | 6.96 | 44.50 | 60.00 | 32.00 | 56.55 | 62.29 | 20.71 | 55.43 | 36.50 |
| GPT-o4-mini | 50.62 | 22.61 | 50.72 | 22.98 | 18.06 | 68.48 | 15.75 | 53.24 | 69.17 | 48.00 | 44.50 | 76.00 | 31.58 | 64.88 | 38.50 |
| **Average** | 35.09 | 15.66 | 36.09 | 15.30 | 20.72 | 40.99 | 6.02 | 31.35 | 29.90 | 17.27 | 35.42 | 40.82 | 15.55 | 35.06 | 24.50 |

CE: Causal effect estimation; CD: Cause-effect direction; CI: Contextual interaction; CF: Counterfactual prediction; DS: Domain shift. MC: Mediator-outcome confounding; NS: Necessity and sufficiency; OE: Observational vs experimental; PS: Population shift; SB: Selection bias. SM: Sequential mediators; SP: Simpson's paradox; SU: Structure uncertainty; TS: Temporal stability; TM: Treatment-mediator interaction.

> **Workflow for Code-Assisted Prompting with Debugging**
>
> 1. **Code Generation:** Same as Protocol 2: provide the causal question, dataset location, column names, and a data sample. Request Python code for analysis.
> 2. **Code Execution:** Extract and run the generated code.
> 3. **Debugging (if execution fails):** Present the error message to the model and request corrected code. Execute the corrected code.
> 4. **Result Interpretation:** Show the model its code and the numerical results, and ask it to interpret the results in context.
> 5. **Output Collection:** Record the model's interpretation for evaluation.

Table 9 compares causal reliability across all three protocols. Debugging primarily benefits models that frequently fail on the first code attempt. For example, Mistral-7B improves from 7.65% to 17.55% and Llama3.1-8B from 12.86% to 19.04%, recovering to or above their direct-prompting baselines. In contrast, stronger models such as GPT-o4-mini and DeepSeek-chat show mild but

consistent improvements, as their initial code-error rates are already low. This pattern suggests that for weaker models, code-assisted prompting introduces a code-generation failure mode that masks their underlying reasoning ability, whereas stronger models leverage computation to improve causal inference.

Table 9: Causal reliability (%) across three evaluation protocols. Causal reliability is the normalized score averaged across all six pitfall categories.

| LLM | Direct | Code-Assisted | Code-Assisted + Debug |
|---|---|---|---|
| Gemma2-9B | 13.69 | 13.24 | 15.93 |
| Llama3.1-8B | 17.86 | 12.86 | 19.04 |
| Llama3.1-70B | 19.78 | 19.89 | 20.42 |
| Mistral-7B | 14.43 | 7.65 | 17.55 |
| Mixtral-8×22B | 16.99 | 20.59 | 20.84 |
| Claude-3.5-Sonnet | 26.22 | 29.38 | 30.17 |
| Gemini-2.0-Flash | 24.36 | 31.80 | 33.30 |
| DeepSeek-Chat | 32.43 | 36.12 | 36.72 |
| GPT-4.1 | 25.24 | 37.32 | 37.98 |
| GPT-o4-Mini | 40.72 | 43.03 | 45.15 |

## I  BENCHMARK COVERAGE COMPARISON

Table 10 compares the challenge coverage of CausalPitfalls against four existing benchmarks. Existing evaluations have focused either on semantic causal reasoning (Jin et al., 2023) or on estimation accuracy in standard settings (Liu et al., 2024; Wang, 2024). In contrast, CausalPitfalls targets common statistical pitfalls and failure modes across all six categories.

Table 10: Coverage of CausalPitfalls challenges across existing benchmarks. "Symbolic only" refers to semantic reasoning without tabular data.

| Challenge | CLadder | Corr2Cause | CausalBench | QRData | Ours |
|---|---|---|---|---|---|
| Simpson's paradox | Symbolic only | No | No | Partial | Yes |
| Selection bias (Berkson's) | Symbolic only | No | No | No | Yes |
| Observational vs experimental | Symbolic only | No | Yes | Partial | Yes |
| Causal effect estimation | Symbolic only | No | Yes | Partial | Yes |
| Counterfactual prediction | Symbolic only | No | Yes | Limited | Yes |
| Causal necessity & sufficiency | Symbolic only | No | Partial | No | Yes |
| Mediator–outcome confounding | Symbolic only | No | Partial | No | Yes |
| Sequential mediators | Symbolic only | No | No | No | Yes |
| Treatment–mediator interaction | No | No | No | No | Yes |
| Cause–effect direction | Symbolic only | Symbolic only | Yes | No | Yes |
| Uncertainty in causal structures | Partial | No | Partial | No | Yes |
| Population shift & transferability | No | No | No | No | Yes |
| Temporal stability | No | No | No | No | Yes |
| Contextual interaction & moderation | No | No | Partial | Partial | Yes |
| Domain shift & transportability | No | No | No | No | Yes |