# OpenReview forum: "Ice Cream Doesn’t Cause Drowning: Benchmarking LLMs Against Statistical Pitfalls in Causal Inference"
_ICLR.cc/2026/Conference — ICLR 2026 Poster_

### Official Review · Reviewer_gZXP · 2025-10-23

**Soundness:** 3
**Presentation:** 2
**Contribution:** 3
**Rating:** 2
**Confidence:** 3

**Summary:**

LLMs have commonly been assessed for causal reasoning and inference tasks; however typically only a subset of these tasks - or plain commonsense reasoning - is evaluated in any benchmark. This paper proposes an exhaustive summary of pitfalls in causal inference to be verified, and proposes an automated pipeline with a LLM to rate answers - this LLM is shown to be in agreement with a large pool of human graders.

**Strengths:**

- Original approach, and first paper I see that extensively reviews all causal inference skills in LLMs. The list of tasks looks exhaustive, and the taxonomy by difficulty looks relevant and consistent with results.

- The automatic grading with gpt-4o is a welcome innovation and seems validated by external graders

**Weaknesses:**

- It is not detailed where exactly the previous evaluations in the literature are missing wrt the challenges in Table 1. A table detailing this would be welcome.

- The process to generate DAGs in l.245-253 is a bit unclear. No precise equations are given. In the sentences, "For every statistical pitfall, we select causal graphs that capture its unique complexities and characteristics. Each challenge is accompanied by five distinct datasets, each containing over 500 samples for comprehensive evaluation.", it is unclear whether five daatsets are generated by graph, or five graphs are chosen by task? Also, the sentence "For every statistical pitfall, we select causal graphs that capture its unique complexities and characteristics. Each challenge is accompanied by five distinct datasets, each containing over 500 samples for comprehensive evaluation." suggests that only linear SCMs are evaluated, which would be restrictive.

- It is unclear how scores are computed exactlty. I understand that points are assigned to questions, but what's the exact assignment? And why assign more than 1 point to each question (scores could then be normalized on a 0-100% scale)? It is also unclear how gpt-4o is prompted to evaluate models, or is evaluation is done exactly with it. Does it choose points itself? Or is it Correct/Incorrect, with Correct giving all points and Incorrect giving none?

**Questions:**

- Can you address/answer the above weaknesses?

- To double check that the discrepancy of 0.11 between gpt-4o and human scores is low, can you also compute it in these cases: a) scores are all zero, b) scores are all maximal, c) scores are uniformly sampled between zero and the maximal possible value (averaged over several seeds).

- Did/Can you test the evaluation pipeline on statistical or causal libraries (eg DoWhy), or on humans?

Note: I am ready to increase my score up to Accept levels if my concerns are addressed.

---

> ### Author Response · Authors · 2025-11-23
> **Rebuttal: Part I**
>
> We appreciate the careful and thoughtful review. We are glad that you found the approach original, that the coverage of causal inference skills and the difficulty-based taxonomy look relevant, and that the GPT-4o-based automatic grading appears well supported by human validation. Below, we address your concerns and questions in detail and will incorporate the corresponding clarifications and updates into the revised manuscript.
>
> >Q: “where exactly the previous evaluations in the literature are missing wrt the challenges in Table 1. A table detailing this would be welcome.”
>
>
> **R**: Thank you for the helpful feedback. We will add the following Table 1 in the revised manuscript and improve the discussion. Specifically, existing evaluations in the literature have focused either on semantic causal reasoning (Jin et al., 2023; Wang, 2024) or on final accuracy in estimating causal effects (Jin et al., 2023; Liu et al., 2024). In contrast, our benchmark is designed to target common pitfalls and failure modes, rather than only evaluating estimation accuracy in the absence of such pitfalls.
>
> >Q:  “The process to generate DAGs in l.245-253 is a bit unclear…suggests that only linear SCMs are evaluated, which would be restrictive...”
>
> **R**: We thank the reviewer for pointing out this ambiguity and for the opportunity to clarify. For each challenge, we first fix a single causal graph (DAG) that encodes the qualitative structure of it (e.g., the confounder–treatment–outcome pattern for Simpson’s paradox). We then generate five distinct datasets from that same DAG: across these datasets, the causal dependencies (graph structure) are identical, but the structural equations and noise realizations differ, such as different coefficients and noise scales. Importantly, the structural equations are not restricted to linear relationships. In the benchmark we allow both linear and non-linear SCMs (e.g. non-linear link functions and interaction terms), so the evaluation is not limited to purely linear models. We will revise the corresponding paragraph to make the data-generation process and the use of non-linear SCMs, and to avoid the impression that only linear relationships are considered.
>
> >Q: “It is unclear how scores are computed exactlty”
>
> **R**: We thank the reviewer for raising this and will clarify the scoring procedure in the revision. For each dataset, we design a pitfall-specific rubric that lists several key requirements (e.g., correctly identifying the pitfall, choosing an appropriate adjustment strategy, interpreting the result correctly). Each rubric item is worth 1 point, and a model’s raw score on that dataset is simply the number of satisfied items divided by the total number of items (e.g., 5 out of 7 items satisfied gives a score of 5). Different pitfalls naturally require different numbers of rubric items, so we always normalize by the maximum possible points, which makes all dataset-level scores comparable on a 0–1 (or 0–100%) scale.
>
> When we use GPT-4o as the automatic judge, it does not choose points or criteria itself. We pass it (i) the model’s answer and (ii) the rubric as an ordered list of criteria, and instruct it to output a 0/1 decision for each criterion. Concretely, the judge is prompted with a template of the following form:
>
> ```bash
> You are a careful grader for causal reasoning answers. Evaluate the following text against these criteria. For each criterion, respond with 1 if the response satisfies the criterion, or 0 if it doesn't.
> CRITERIA:
> {criteria_as_JSON}
> Response TO EVALUATE:
> {model_answer}
> Return ONLY a JSON object with an array of binary scores (0 or 1) in the same order as the criteria.
> ```
> We then compute the normalized score as the mean of this 0-1 vector. We will add this description, together with an example rubric and judge prompt, to the revised manuscript.
>
> >Q: “To double check that the discrepancy of 0.11 between gpt-4o and human scores is low, can you also compute it in these cases…”
>
> **R**: We thank the reviewer for this comment and for pointing out the concern. We realized this confusion arises because the definition of the discrepancy metric was not stated clearly enough. Specifically, for each response $i$, we compute normalized scores $ns_{\text{GPT-4o},i}$  and $ns_{\text{human},i}$, taking values in $[0,1]$. The discrepancy is then computed as the average of $$|ns_{\text{GPT-4o},i}-ns_{\text{human},i}|.$$ By construction, this quantity is between 0 and 1, with smaller values indicating closer agreement between human and GPT-4o scores.
>
> Under this definition, in cases (a) and (b) (in our understanding), where human and GPT-4o scores are always identical (all zero or all maximal for both), the discrepancy is 0. In contrast, if one grader always gives the minimum (0) and the other always gives the maximum, the discrepancy is 1.

---

> ### Author Response · Authors · 2025-11-23
> **Rebuttal: Part II**
>
> >Q: Did/Can you test the evaluation pipeline on statistical or causal libraries (eg DoWhy), or on humans?”
>
> **R**: We thank the reviewer for this helpful suggestion. Libraries such as DoWhy are not end-to-end analysts: they expose functions that require a human or an LLM to specify the causal model, choose estimators, and interpret the results, rather than automatically producing full causal responses. In our code-assisted protocol, LLMs already call functions in causal libraries as part of their analysis. Specifically, in practice, there’s frequent use of standard statistical and causal libraries, including DoWhy (e.g., statsmodels, linearmodels, dowhy, econml, causalml, causalnex, causalimpact). We will follow this comment and improve the explanations.
>
>
>
> **Table 1: Coverage of CausalPitfalls challenges across existing benchmarks**
> *“Yes/No/Partial” indicate whether a benchmark  covers the corresponding challenge. “Symbolic only” refers to semantic reasoning without tabular data.*
>
>
> | **CausalPitfalls Challenge**                          | **CLadder**           | **Corr2Cause** | **CausalBench (Wang, 2024)**        | **QRData** | **CausalPitfalls (Ours)** |
> |------------------------------------------------------|-----------------------|----------------|-------------------------------------|-----------|---------------------------|
> | Simpson’s paradox                                    | Symbolic only         | No             | No                                  | Partial   | **Yes**                   |
> | Selection bias (Berkson’s / collider bias)           | Symbolic only         | No             | No                                  | No        | **Yes**                   |
> | Observational vs experimental reasoning              | Symbolic only         | No             | Yes (Text/Code)                     | Partial   | **Yes**                   |
> | Causal effect estimation                             | Symbolic only         | No             | Yes (Text/Code)                     | Partial   | **Yes**                   |
> | Counterfactual outcome prediction                    | Symbolic only         | No             | Yes (Text/Code)                     | Limited   | **Yes**                   |
> | Causal necessity and sufficiency                     | Symbolic only         | No             | Partial (logical/semantic)          | No        | **Yes**                   |
> | Mediator–outcome confounding                         | Symbolic only         | No             | Partial (Front-door)                | No        | **Yes**                   |
> | Sequential mediators                                 | Symbolic only         | No             | No                                  | No        | **Yes**                   |
> | Treatment–mediator interaction effects               | No                    | No             | No                                  | No        | **Yes**                   |
> | Cause–effect direction inference                     | Symbolic only         | Symbolic only  | Yes                                 | No        | **Yes**                   |
> | Handling uncertainty in causal structures            | Partial (graph classes) | No           | Partial                             | No        | **Yes**                   |
> | Population shift and transferability                 | No                    | No             | No                                  | No        | **Yes**                   |
> | Temporal stability of causal effects                 | No                    | No             | No                                  | No        | **Yes**                   |
> | Contextual interaction and moderation                | No                    | No             | Partial                             | Partial   | **Yes**                   |
> | Domain shift and transportability                    | No                    | No             | No                                  | No        | **Yes**                   |
>
> **References**:
>
> Jin, Zhijing, et al. "Cladder: Assessing causal reasoning in language models." NeurIPS, 36 (2023): 31038-31065.
>
> Jin, Zhijing, et al. "Can large language models infer causation from correlation?." arXiv preprint arXiv:2306.05836 (2023).
>
> Wang, Zeyu. "Causalbench: A comprehensive benchmark for evaluating causal reasoning capabilities of large language models." in SIGHAN-10. 2024.
>
> Liu, Xiao, et al. "Are llms capable of data-based statistical and causal reasoning? benchmarking advanced quantitative reasoning with data." arXiv preprint arXiv:2402.17644 (2024).

---

> > ### Comment · Reviewer_gZXP · 2025-11-25
> >
> > Many thanks for the helpful rebuttal, it has alleviated all my concerns! I will update my score to a clear Accept.

---

> > > ### Author Response · Authors · 2025-11-25
> > >
> > > Thank you for your quick response and raising the score to accept! We are happy that we have addressed your concerns and will incorporate them in the revision. Thank you again for your effort in reviewing our paper.

---

### Official Review · Reviewer_iHLz · 2025-10-31

**Soundness:** 4
**Presentation:** 4
**Contribution:** 3
**Rating:** 8
**Confidence:** 5

**Summary:**

This paper introduces CausalPitfalls, a benchmark to test whether LLMs can do statistical causal inference without falling into classic traps like Simpson’s paradox, selection/collider bias, et.c. It shows that current models often give confident but statistically wrong causal conclusions that are highly sensitive to phrasing, and that letting them run code helps but doesn’t solve the problem overall.

Contributions:

* 6 causal-pitfall families, 15 challenges, 75 questions + 75 SCM-based datasets, each with 5 difficulty levels, targeting: confounding/spurious association, interventions vs observation, counterfactuals, mediation, causal discovery, and external validity/transport.
* (1) Direct prompting to measure intrinsic causal reasoning from raw data; (2) Code-assisted prompting where the model generates and runs statistical code before answering. This separates questions of “can the model spot the pitfall?” and “can the model reason its way out of it?”
* A rubric-based, normalized metric for causal reliability validated against human statisticians to compare models on trustworthiness, not just accuracy.
* Illustrative failure cases including models flipping answers based on branding (“HealthPlus” vs “UltraSugar”) on identical data; falsely detecting gender bias / Simpson’s paradox with age stratification in synthetic medical data, falsely treating Berkson’s/collider bias in hospital data as a real effect, over-attributing effects in mediator–outcome confounding setups, fail to detect non-identifiability/Markov equivalence in causal discovery.
* Empirically findings: the best model reaches only ~40–43% causal reliability, mediation and external validity are especially difficult

**Strengths:**

In my view, the main strength of the paper is demonstrating cases where the LLM fails to resolve common pitfalls in causal inference across a broad set of causal inference problems, using generative causal graphical model ground truth. The paper then extends this analysis into a formal benchmark with a useful metric focused on evaluating LLMs for how reliable they are in answering these types of questions.

**Weaknesses:**

The benchmark relies heavily of the LLMs ability to parse raw data.

**Questions:**

Could you have converted the raw data into sufficient statistics sufficient to resolve the causal inference problems (e.g., counts, proportions, rates, descriptions that imply monotonicity in case of probabilities of causation, etc.), then construct prompts based on these sufficient stats? One could do this without giving cues to the LLM about the right way to solve a problem (e.g., providing additional summary statistics that not needed to correctly do the inference). This would have better evaluated a models' ability to "talk" its way through the inference problem and less dependent on its ability to parse raw data.

---

> ### Author Response · Authors · 2025-11-23
> **Rebuttal**
>
> We appreciate your careful and detailed review. We are glad that you see value in using SCM-based ground truth to expose LLM failures on common causal pitfalls and in turning this into a formal benchmark with a rubric-based causal reliability metric. Below, we address your concerns and will incorporate the corresponding changes into the revised manuscript.
>
> >Q: “The benchmark relies heavily of the LLMs ability to parse raw data” “...converted the raw data into sufficient statistics sufficient to resolve the causal inference problems…”
>
> **R**: We thank the reviewer for this insightful suggestion. We agree that inference based on sufficient statistics, rather than full raw tables, can better evaluate an LLM’s causal reasoning (independent of low-level data parsing). We will expand our discussion of this point in the revised manuscript.
>
> We also want to clarify that our code-assisted protocol already involves a data-to-statistics conversion step. In this setting, the LLM is given the column names and a small sample of rows, and is asked to write Python code that performs the analysis it considers appropriate. We then execute this code and feed the resulting numerical outputs back to the model, which must interpret them to answer the causal question. In other words, the model itself decides how raw data should be transformed into the relevant summary statistics. Therefore, the benchmark evaluates both whether it can design a correct analysis pipeline (choose and implement suitable transformations) and whether, once those statistics are available, it can reason correctly about the underlying causal pitfall.
>
> During the revision phase, we also introduced a new variant, Code-Assisted Prompting with Debugging: when the initially generated code fails, we show the error message to the model and allow a single attempt to correct the code. As reported in Table 1 of this response, this variant yields higher causal reliability, especially for models that previously had high code-error rates (Figure 4 in the manuscript). \textbf{Taken together, these results show that direct prompting, which relies entirely on raw-table reasoning, performs worse than both code-assisted protocols, which instead use code to convert raw data into statistics and then analyze those statistics.} This aligns with the reviewer’s insightful observation that evaluation based on summarized information (statistics) is often more indicative of an LLM’s true causal reasoning ability than evaluation based only on raw row-level parsing.
>
> **Table 1: Causal reliability (%) across three evaluation protocols**
> *Causal reliability is defined in the paper as the normalized score averaged across the six pitfall categories: Confounding, Interventions, Counterfactuals, Mediation, Causal Discovery, and External Validity.*
>
>
> | **Protocol**               | **Gemma2-9B** | **Llama3.1-8B** | **Llama3.1-70B** | **Mistral-7B** | **Mixtral-8×22B** | **Claude-3.5-Sonnet** | **Gemini-2.0-Flash** | **DeepSeek-Chat** | **GPT-4.1** | **GPT-o4-Mini** |
> |----------------------------|---------------|------------------|-------------------|----------------|--------------------|------------------------|------------------------|--------------------|-------------|------------------|
> | **Direct Prompting**       | 13.69        | 17.86           | 19.78            | 14.43         | 16.99             | 26.22                 | 24.36                 | 32.43             | 25.24      | 40.72           |
> | **Code-Assisted Prompting**| 13.24        | 12.86           | 19.89            | 7.65          | 20.59             | 29.38                 | 31.80                 | 36.12             | 37.32      | 43.03           |
> | **Code-Assisted (with debugging)**  | 15.93      | 19.04         | 20.42            | 17.55          | 20.84            |  30.17            | 33.30      | 36.72      | 37.98   | 45.15     |

---

### Official Review · Reviewer_Z3sS · 2025-11-01

**Soundness:** 3
**Presentation:** 3
**Contribution:** 3
**Rating:** 4
**Confidence:** 4

**Summary:**

The paper introduces CausalPitfalls, a benchmark designed to evaluate the capability of LLMs in recognizing and thus overcoming common causal inference pitfalls. The authors claim this will improve LLMs' capability of handling statistical causal inference. They show this by getting LLM generated text and code evaluated by human experts.

**Strengths:**

1. Advancing causal inference research is a welcome topic in the field of LLMs
2. The structure of the benchmark and the design aspect of it seems well-thought out and illustrated
3. Having human evaluation for LLM generated content is key

**Weaknesses:**

1. The datasets are at the end of the day still synthetic and not drawn from real observational/experimental studies
2. It is not clear how well this will generalize to real world scenarios
3. It would be helpful if the authors expanded a bit more on what is an acceptable success rate at these tasks. How well do humans do?

**Questions:**

1. Do the authors fear that benchmarks for LLMs could be a losing cause? For example, is it possible that this benchmark becomes "training data" - in other words can LLMs be gamed to do well on benchmarks but actually still be bad at causal inference?
2. Having said that, do the authors have any plans to extend their work to include more types of pitfalls in the future?
3. What if you tell LLMs that they were wrong and they should try again. Do you expect these second chance answers to improve?

---

> ### Author Response · Authors · 2025-11-23
> **Rebuttal: Part I**
>
> We thank Reviewer Z3sS for the thoughtful assessment. We are glad that you see advancing causal inference with LLMs as a valuable direction and that you found the benchmark design and use of human evaluation important. Below, we address your concerns and will include them in the revision.
>
>
> > Q: “synthetic and not drawn from real observational/experimental studies…”
>
> **R**: We thank the reviewer for the thoughtful comments on synthetic data and generalization. Our decision to base CausalPitfalls on structural causal models is intentional: for each pitfall family we specify different DAG and structural equations, so both the causal estimands and the correct handling of the pitfall are therefore determined. In typical observational datasets, even well-known ones, different analysts adopt different causal assumptions and can reach conflicting conclusions; in that setting it is difficult for a benchmark context to say whether a model is objectively right or wrong. The simulated SCMs give us ground truth and allow us to evaluate causal reliability in a principled way, rather than on agreement with any particular analyst.
>
>
> >Q: “not clear how well this will generalize to real world scenarios”
>
>
> **R**: Thank you for the concern on the real-world scenarios. We hope to clarify that these pitfalls are not artificial toy problems but issues that practitioners worry about in real analyses (Rosenbaum 2002). Our goal with CausalPitfalls is not to reproduce every complexity of real datasets, but to provide a clean benchmark for how well LLMs handle these specific, well-known failure modes when the underlying causal structure is known. We appreciate this feedback, and we will include discussions in the revision.
>
>
> >Q: “expanded a bit more on what is an acceptable success rate at these tasks. How well do humans do?”
>
> **R**: We appreciate this question and agree we should include more discussions. The causal reliability score is built from a small set of rubric items, each corresponding to a specific textbook-level mistake (for example, conditioning on a collider, failing to adjust for an obvious confounder, or misreading a classic Simpson’s paradox reversal). The benchmark is intentionally focused on these standard pitfalls, where a well-trained, careful statistician is expected to satisfy essentially all rubric items. In this sense, losing points indicates that the model has committed at least one of these standard conceptual errors. In the revision, we will explain more about the choice of rubrics and how LLMs and experts are supposed to perform.
>
>
>
>
>
> >Q: “Do the authors fear that benchmarks for LLMs could be a losing cause?...” ”...do the authors have any plans to extend their work to include more types of pitfalls in the future?”
>
>
> **R**: We appreciate this thoughtful question and agree that many existing benchmarks have ended up in training sets. In CausalPitfalls, each pitfall family corresponds to a specific type of failure mode and can be instantiated by many different structural causal models (SCMs). From these SCMs, we can generate multiple datasets and question variants by varying parameters, sample sizes, and narratives, so there is no single fixed dataset or prompt to memorize. Many of our tasks also ask the model to design and interpret a full analysis (often via code), rather than just produce a fixed answer. Thus, doing well requires implementing the right statistical operations rather than memorization. We plan to extend the number of SCMs for each challenge, and include more pitfalls in the future.
>
> **References**
>
> Rosenbaum, Paul R. "Observational studies." Observational studies. New York, NY: Springer, New York, 2002. 1-17.

---

> ### Author Response · Authors · 2025-11-23
> **Rebuttal Part II**
>
> >Q: “What if you tell LLMs that they were wrong and they should try again. Do you expect these second chance answers to improve?”
>
>
> **R**: We thank the reviewer for raising this point. We agree that giving models feedback and a second chance to revise their answer is an important future direction, especially in settings where humans and LLMs work together on an analysis. In this paper, we did not reveal the underlying ground truth to LLM to get the causal response.
> Motivated by this suggestion, we have conducted a new protocol, code-assisted prompting with debugging. In this setting, the model first generates code and we execute it on the data; if execution fails, we return the error message and LLM gives revised code.  LLM then produces causal responses based on the execution results of revised code. As shown in the following  Table 1, with a second chance to fix failed code, LLM consistently has improved performance over the code-assisted protocol.
>
> **Table 1: Causal reliability (%) across three evaluation protocols**
> *Causal reliability is defined in the paper as the normalized score averaged across the six pitfall categories: Confounding, Interventions, Counterfactuals, Mediation, Causal Discovery, and External Validity.*
>
>
> | **Protocol**               | **Gemma2-9B** | **Llama3.1-8B** | **Llama3.1-70B** | **Mistral-7B** | **Mixtral-8×22B** | **Claude-3.5-Sonnet** | **Gemini-2.0-Flash** | **DeepSeek-Chat** | **GPT-4.1** | **GPT-o4-Mini** |
> |----------------------------|---------------|------------------|-------------------|----------------|--------------------|------------------------|------------------------|--------------------|-------------|------------------|
> | **Direct Prompting**       | 13.69        | 17.86           | 19.78            | 14.43         | 16.99             | 26.22                 | 24.36                 | 32.43             | 25.24      | 40.72           |
> | **Code-Assisted Prompting**| 13.24        | 12.86           | 19.89            | 7.65          | 20.59             | 29.38                 | 31.80                 | 36.12             | 37.32      | 43.03           |
> | **Code-Assisted (with debugging)**  | 15.93      | 19.04         | 20.42            | 17.55          | 20.84            |  30.17            | 33.30      | 36.72      | 37.98   | 45.15     |

---

### Official Review · Reviewer_fhn1 · 2025-11-01

**Soundness:** 3
**Presentation:** 3
**Contribution:** 3
**Rating:** 6
**Confidence:** 4

**Summary:**

The paper introduces CausalPitfalls, a benchmark designed to evaluate large language models (LLMs) on statistical causal inference tasks while specifically testing their susceptibility to classical causal reasoning errors. The benchmark comprises 75 datasets and 15 challenges across six categories, including confounding, interventions, counterfactual reasoning, mediation, causal discovery, and external validity. It assesses two protocols, direct prompting and code-assisted prompting, and proposes a quantitative metric called causal reliability to standardize comparison across models. Results show that all evaluated LLMs exhibit systematic reliability gaps, particularly on tasks involving mediation and generalization, with moderate improvements observed when code execution is permitted.

**Strengths:**

The paper makes a timely and significant contribution by systematically targeting the reliability rather than accuracy of LLMs in causal inference, an aspect underexplored in previous benchmarks such as CausalBench and CLADDER. The structured coverage of classical statistical pitfalls like Simpson’s and Berkson’s paradoxes, mediation confounding, and counterfactual reasoning reflects a solid understanding of the causal inference literature and provides a rigorous stress test for LLMs. The benchmark’s dual evaluation protocol is well-motivated and demonstrates a clear methodological insight: executable statistical reasoning through code-assisted prompting substantially mitigates causal errors that stem from linguistic biases. The proposed causal reliability metric is also a valuable addition that enables consistent quantitative comparisons across diverse causal reasoning challenges.

**Weaknesses:**

Although overall a good contribution, The benchmark lacks a bit in scope and interpretation. Although the benchmark convincingly demonstrates single LLM limitations, the current coverage only includes very basic prompting stratagies. Current causal inference methods don't use LLMs directly anyways, benchmarking more architectures like finetuned models for causal inference, or better prompting stratagies like React would be more helpful. Moreover, while the paper positions code-assisted prompting as a key improvement, it does not sufficiently analyze why certain models benefit more than others or what kinds of causal reasoning errors persist even after code execution. Without this diagnostic breakdown, the benchmark feels more descriptive than explanatory.

**Questions:**

Refer to the weaknesses section above, additionally, How prevalent are the main pitfalls highlighted in the benchmark within real-world datasets, and to what extent do they typically affect practical causal analyses?

---

> ### Author Response · Authors · 2025-11-23
> **Rebuttal: Part I**
>
> We thank the reviewer for their thoughtful and positive assessment of CausalPitfalls as a "timely and significant contribution'', and for highlighting our focus on reliability (rather than accuracy) and the structured coverage of classical pitfalls.
>
> We also appreciate your suggestions to broaden the evaluated LLM configurations, to deepen the diagnostic analysis of code-assisted prompting, and to clarify the real-world prevalence and impact of the pitfalls. We address each of these points in turn below.
>
> >Q: benchmarking more architectures like finetuned models for causal inference, or better prompting stratagies like React would be more helpful.”
>
> **R**: Thank you for the suggestion. We agree that evaluating finetuned models or stronger prompting strategies is valuable. We would like to clarify that our code-assisted setting already follows the same core idea as ReAct: the model inspects sample data (Appendix C.1), writes Python code to compute the relevant statistics, we execute the code, and the LLM then produces its final response.
>
> In addition, following the reviewer’s insightful comment, we introduced a new variant: Code-Assisted Prompting with Debugging. When the generated code fails, we show the LLM the error message and allow one opportunity to correct the code. The results are shown in Table 1. This leads to higher causal reliability, especially for models that previously had high code-error rates.
>
>
> **Table 1: Causal reliability (%) across three evaluation protocols**
> *Causal reliability is defined in the paper as the normalized score averaged across the six pitfall categories: Confounding, Interventions, Counterfactuals, Mediation, Causal Discovery, and External Validity.*
>
>
> | **Protocol**               | **Gemma2-9B** | **Llama3.1-8B** | **Llama3.1-70B** | **Mistral-7B** | **Mixtral-8×22B** | **Claude-3.5-Sonnet** | **Gemini-2.0-Flash** | **DeepSeek-Chat** | **GPT-4.1** | **GPT-o4-Mini** |
> |----------------------------|---------------|------------------|-------------------|----------------|--------------------|------------------------|------------------------|--------------------|-------------|------------------|
> | **Direct Prompting**       | 13.69        | 17.86           | 19.78            | 14.43         | 16.99             | 26.22                 | 24.36                 | 32.43             | 25.24      | 40.72           |
> | **Code-Assisted Prompting**| 13.24        | 12.86           | 19.89            | 7.65          | 20.59             | 29.38                 | 31.80                 | 36.12             | 37.32      | 43.03           |
> | **Code-Assisted (with debugging)**  | 15.93      | 19.04         | 20.42            | 17.55          | 20.84            |  30.17            | 33.30      | 36.72      | 37.98   | 45.15     |
>
>
> Specifically, debugging mainly benefits models that often fail on the first code attempt (such as Mistral-7B and Llama-8B), while larger models show mild but consistent improvements.
>
> We also checked for LLMs finetuned for statistical causal inference to include, but we couldn’t find any suitable open-source options. If the reviewer knows of one, we would be happy to add it.

---

> ### Author Response · Authors · 2025-11-23
> **Rebuttal: Part II**
>
> >Q: “...it does not sufficiently analyze why certain models benefit more than others or what kinds of causal reasoning errors persist even after code execution…”
>
>
> **R**: Thank you for raising this point. We agree that a clearer diagnostic breakdown is helpful, and we will include this in the revision. At the model level, our results already show that some LLMs benefit strongly from code assistance while others do not: for example, GPT-4.1 and Gemini-2.0-flash improve substantially under code-assisted prompting (Table 4), whereas Llama3.1-8B and Mistral-7B perform worse with code but recover once we allow a single debugging step (Table 1 in the above response). This pattern indicates that the latter group is mainly limited by unreliable code generation, while the former can produce reasonable code.
>
> >Q: “How prevalent are the main pitfalls highlighted in the benchmark within real-world datasets, and to what extent do they typically affect practical causal analyses?”
>
>
> **R**: The pitfalls we model in CausalPitfalls are repeatedly described as routine threats in observational causal analyses (Rosenbaum, 2002). For example, major texts emphasize that some unmeasured confounding is essentially unavoidable in observational data, and much of modern causal methodology is devoted to handling it. Selection and collider bias have likewise been documented as common sources of substantial distortion in epidemiologic and clinical studies (Hernán, 2004). Mediation analyses are now widely used across epidemiology, psychology, and the social sciences, with a large literature showing that modest violations of mediator–outcome assumptions can meaningfully bias direct and indirect effect estimates (VanderWeele, 2016).
>
> In practice, these pitfalls can do much more than introduce small error: they can (i) reverse the qualitative conclusion of a study by flipping the sign of an estimated effect (as in Simpson’s paradox), (ii) produce apparently strong, statistically significant associations where the true causal effect is essentially zero (collider/selection bias), and (iii) lead to effect estimates that look convincing in the original sample but fail badly when the intervention is deployed in a different hospital, region, or time period (external-validity failures).
>
> The pitfalls focus on standard problems in applied causal inference that are known to have severe consequences, including changing not just the size but sometimes even the sign of practical causal conclusions.
>
> ## References
>
> Rosenbaum, Paul R. "Observational studies." Observational studies. New York, NY: Springer New York, 2002. 1-17.
>
> Hernán, Miguel A., Sonia Hernández-Díaz, and James M. Robins. "A structural approach to selection bias." Epidemiology 15.5 (2004): 615-625.
>
> VanderWeele, Tyler J. "Mediation analysis: a practitioner's guide." Annual review of public health 37.1 (2016): 17-32.

---

### Author Response · Authors · 2025-11-23
**Rebuttal Summary**

We thank all reviewers for their time, careful reading, and constructive feedback. We are encouraged by the positive comments on the originality of the benchmark, the structured coverage of classical causal pitfalls using structural causal model (SCM)–based ground truth, and the rubric-based evaluation with human validation and GPT-4o as an automatic judge.

In response to the reviews, we
1. introduce and report results for a new “code-assisted with debugging” protocol across all challenges, which improves causal reliability, especially for LLMs with high code-error rates;
2. clarify the rationale for using synthetic SCM-based data, the directed acyclic graph (DAG) and dataset generation process (including the use of non-linear SCMs), and the real-world prevalence and impact of the pitfalls we study;
3. make the scoring scheme, rubric design, GPT-4o judging prompt, and the definition and interpretation of the human-GPT-4o discrepancy metric clear;
4. add a comparison table showing how CausalPitfalls complements existing causal reasoning benchmarks;
5. expand the discussion of how to interpret causal reliability scores and what level of performance would be expected from a careful human analyst.

We will incorporate these clarifications, tables, and new experimental results into the revised manuscript.

---

### Comment · Area_Chair_RCht · 2025-11-27

Dear Authors and Reviewers,

The discussion phase will end soon. If you want to further discuss comments and replies with each other, please post your thoughts by adding official comments.

Thanks for your efforts and contributions to ICLR 2026.

Best regards,

Your Area Chair

---

### Meta-Review · Area_Chair_Lack · 2025-12-29

**Summary:**

This paper presents a new benchmark named CausalPitfalls, which is designed to evaluate the capability of large language models (LLMs) in overcoming common causal inference pitfalls. This new benchmark contains 75 datasets and 15 challenges across six categories, i.e., confounding, interventions, counterfactual reasoning, mediation, causal discovery, and external validity. The benchmark involves two protocols, including direct prompting and code-assisted prompting. In addition, the authors proposed a quantitative metric named Causal Reliability to standardize comparison across models. Results and discussions are reported in the paper.

Reviewers agreed that this paper makes a timely and significant contribution by targeting the reliability rather than accuracy of LLMs in causal inference. The benchmark’s evaluation protocol is well-motivated and clearly demonstrated. Moreover, human evaluations included in the paper are meaningful.

Meanwhile, reviewers raised some concerns regarding the benchmark's scope and interpretation, comprehensive analysis of results, generalization to real-world scenarios, evaluation protocols, technical details, etc.

**Reviewer Concerns:**

The authors have provided very detailed responses to address the concerns from reviewers. In particular, a new “code-assisted with debugging” protocol has been added to further evaluate causal reliability. A new table that describes how CausalPitfalls complements existing causal reasoning benchmarks has been added to the revised paper. In addition, details about the scoring scheme, rubric design, GPT-4o judging prompt, and the definition and interpretation of the human-GPT-4o discrepancy metric are also clarified in the response and the revised paper. Most of the reviewers' prior concerns have been well addressed by the rebuttal.

**Reviewer Scores:**

Initially, this paper received diverse ratings: 8, 6, 4, and 2. During the rebuttal and discussion period, the reviewer with rating 2 explicitly mentioned that they will update the score to a clear Accept. Considering this evidence as well as the overall good quality of the authors' rebuttal, in my opinion, it is very likely that the final overall scores will be positive.

---

### Decision · Program_Chairs · 2026-01-26

Accept (Poster)